# Ultra-permeable silk-based polymeric membranes for vacuum-driven nanofiltration

Bowen Gan[1], Lu Elfa Peng[1], Wenyu Liu[1], Lingyue Zhang[1], Li Ares Wang[1], Li Long[1], Hao Guo[2], Xiaoxiao Song[3], Zhe Yang[1,4] ✉ & Chuyang Y. Tang[1] ✉

Nanofiltration (NF) membranes are commonly supplied in spiral-wound modules, resulting in numerous drawbacks for practical applications (e.g., high operating pressure/pressure drop/costs). Vacuum-driven NF could be a promising and low-cost alternative by utilizing simple components and operating under an ultra-low vacuum pressure (<1 bar). Nevertheless, existing commercial membranes are incapable of achieving practically relevant water flux in such a system. Herein, we fabricated a silk-based membrane with a crumpled and defect-free rejection layer, showing water permeance of $96.2 \pm 10$ L m$^{-2}$ h$^{-1}$ bar$^{-1}$ and a Na$_2$SO$_4$ rejection of $96.0 \pm 0.6\%$ under cross-flow filtration mode. In a vacuum-driven system, the membrane demonstrates a water flux of $56.8 \pm 7.1$ L m$^{-2}$ h$^{-1}$ at a suction pressure of 0.9 bar and high removal rate against various contaminants. Through analysis, silk-based ultra-permeable membranes may offer close to 80% reduction in specific energy consumption and greenhouse gas emissions compared to a commercial benchmark, holding great promise for advancing a more energy-efficient and greener water treatment process and paving the avenue for practical application in real industrial settings.

Nanofiltration (NF) membranes have pores of 0.5–2 nm and charged membrane surfaces[1–3], enabling them for selective retention/passage of small molecules and ions. With excellent selectivity for water/solute and solute/solute, NF technology has been widely used in various applications for treating diverse water types (e.g., wastewater, groundwater, and saline water)[4] and recovering resources[5]. In practical applications, NF membranes are commonly supplied in spiral-wound modules and installed in pressure vessels, where a high transmembrane pressure (TMP) of up to 10 bar is applied to drive the separation process[6]. Unfortunately, this configuration often suffers from inevitable problems, such as high-pressure drop[7], spacer-induced membrane fouling[8], and costly pressure vessels[9]. These drawbacks bring out high energy consumption and additional equipment expenses, which further limits the application scenarios of NF. Therefore, alternative module designs are required to reform NF technologies.

A promising alternative is a submerged vacuum-driven NF, where the feed water is directly drawn through membranes by a suction force[10]. Historically, submerged microfiltration (MF) and ultrafiltration (UF) membrane modules have been extensively employed in water-related applications with great success, notably in membrane bioreactors (MBRs) that revolutionized wastewater treatment[11–13]. Compared with traditional porous MF/UF-based processes[14], submerged NF membranes could provide a high rejection of a wide range of contaminants such as nutrients, heavy metals, and micropollutants. Nevertheless, practical applications of submerged NF operation have been rarely reported up to date[14,15]. This operation is critically

[1]Department of Civil Engineering, The University of Hong Kong, Pokfulam, Hong Kong SAR, China. [2]Institute of Environment and Ecology, Shenzhen International Graduate School, Tsinghua University, Shenzhen, China. [3]Centre for Membrane and Water Science and Technology, Ocean College, Zhejiang University of Technology, Hangzhou, China. [4]Dow Centre for Sustainable Engineering Innovation, School of Chemical Engineering, The University of Queensland, Brisbane QLD 4072, Australia. ✉e-mail: zheyang@connect.hku.hk; tangc@hku.hk

constrained by a maximum suction pressure of 1 bar, such that existing commercial NF membranes (with low water permeance on the order of 10 L m$^{-2}$ h$^{-1}$ bar$^{-1}$) are generally incapable of achieving practically relevant water fluxes. This research gap can be addressed by developing ultra-permeable NF membranes (e.g. with water permeance approaching 100 L m$^{-2}$ h$^{-1}$ bar$^{-1}$). Such ultra-permeable NF membranes would translate into ultra-low energy consumption (by operating under partial vacuum), reduced cost and footprint (by eliminating pressure vessels), and easy retrofitting to existing treatment works, leading to opportunities for applications[6].

Numerous efforts have been dedicated to enhancing NF water permeance[16–23], but ultra-permeable NF performance (on the order of 100 L m$^{-2}$ h$^{-1}$ bar$^{-1}$) has seldom been achieved. NF membranes, commonly with a thin-film composite (TFC) structure, are prepared by forming a polyamide (PA) rejection layer via interfacial polymerization (IP) on a porous UF substrate (e.g., polysulfone and polyethersulfone membranes). However, conventional TFC membranes have faced a critical challenge of the funnel effect. For water to reach substrate pores, its transport distance inside the PA rejection layer is much greater than the thickness of the layer (due to the additional transverse distance[24]), resulting in orders of magnitude increase in transmembrane hydraulic resistance and thus insufficient water permeance[25,26]. As a result, traditional UF substrates with low surface porosity (often <10%) severely constrain NF water permeance. To fabricate ultra-permeable NF membranes, we construct a biometric silk nanofiber (SNF)-coated substrate with high porosity and rough surface. Indeed, SNF has been widely used in various applications, including water and air filter[27], osmotic energy harvest[28], drug sustained release[29], biological scaffolds[30], and sensors[31]. The broad applications rely on the distinguished characteristics of SNF, such as excellent biological compatibility, high mechanical strength, suitable dimension in nanoscale, and well-dispersibility in water[32,33]. In the context of NF membrane fabrication, the peptide bonds and charged amino acid residues in SNF are compatible with PA chemistry and could potentially improve the interaction with piperazine (PIP) monomers during the IP process. These prominent features prompt us to introduce SNF nanomaterials into thin and porous coating layers on the rough substrates to induce favorable formation conditions for PA rejection layer.

Herein, we developed a substrate-templated method to fabricate a defect-free ultra-permeable NF membrane by introducing SNF. Before the IP reaction, we preloaded thin and porous SNF coating layers on a rough MF substrate surface (Supplementary Fig. 1). Compared with unmodified MF substrate forming a defective PA layer (Fig. 1A), the SNF-coated substrate could simultaneously mitigate the funnel effect and grow a crumpled yet intact PA layer with an enlarged larger filtration area (Fig. 1B). The resulting SNF-incorporated NF membrane (SNF-NF) exhibited high permeance of 96.2 L m$^{-2}$ h$^{-1}$ bar$^{-1}$ (nearly 10-fold higher than commercial NF membranes) while maintaining a high Na$_2$SO$_4$ rejection of 96.0% under cross-flow filtration. We further demonstrated this ultra-permeable membrane in a vacuum-driven NF system, simultaneously achieving an outstanding water flux of 56.8 L m$^{-2}$ h$^{-1}$, Na$_2$SO$_4$ rejection of 96.3%, and perfluorooctane sulfonate (PFOS) removal of 99.6% at a partial vacuum of 0.9 bar. Compared to a commercial benchmark NF270, our SNF-NF membranes in vacuum-driven NF systems enable a nearly 80% reduction in specific energy consumption (SEC) and greenhouse gas (GHG) emissions, holding great promise for advancing sustainable water treatment and providing a paradigm shift towards a more energy-efficient and greener application of NF technology.

## Results and discussion
### Characterization of SNF
Before fabricating the SNF-NF membrane, we first prepared and characterized SNF. SNF is a type of natural protein polymer, which can

be directly extracted from silkworm silks via simple physical-chemical processing (Supplementary Fig. 2). Supplementary Fig. 3A–E shows that SNF has excellent dispersibility in water and nano-sized dimensions (9.0 ± 2.7 nm in width and 380 ± 200 nm in length). In addition, flourier transform infrared spectroscopy (FTIR, Supplementary Fig. 3F) indicates SNF contained abundant β-sheet structures with absorbance peaks (around 1514, 1618, and 1647 cm$^{-1}$) ascribed to amide band regions[28]. This structure can contribute to the excellent stability of SNF film due to the strong hydrogen bonds and van der Waals forces[28,34]. These properties of SNF could potentially avoid the agglomeration of nanomaterials and contribute to forming a stable homogeneous coating layer on substrates (the stability characterization for SNF coatings, Supplementary Fig. 4, 5).

### Morphology of SNF-NF membranes
We further examined the physicochemical properties of the control (SNF0-NF0.5 without SNF loading) and an SNF-NF membrane loaded with 20 mL of 129 µg mL$^{-1}$ SNF (SNF20-NF0.5). As shown in the SEM plane and cross-sectional views in Fig. 1C, D, both SNF0-NF0.5 and SNF20-NF0.5 inherited the rough features of MF substrates. Specifically, both NF membranes present a $Ra$ ~ or > 200 nm (Fig. 1C-3, D-3), which is more than one order of magnitude higher than conventional commercial NF membranes (often in the range of 10–20 nm[19,35]). At the same time, their roughness values are comparable to that of the substrate ($Ra$ = 388 nm, Supplementary Fig. 6A). Interestingly, we observed some defects (pinholes) in the SNF0-NF0.5 membrane (Fig. 1C-1, and more details in Supplementary Fig. 7), which can be ascribed to the rough MF substrate that interfered with the distribution of PIP solution during the IP process[36]. In addition, due to the large pore size of the MF substrate (mean pore size of 0.218 µm, Supplementary Fig. 8), the formed thin PA rejection layer has to span over the large pore region (Supplementary Fig. 6A), resulting in poor mechanical stability and defect formation[37]. In contrast, SNF20-NF0.5 showed a defect-free surface (Fig. 1D-1), which could be attributed to the excellent compatibility of SNF with PA chemistry. Specifically, silk fibers feature similar chemistry to the PA rejection layer−both materials have amide functional groups that ensure good compatibility[38]. Furthermore, the hydrophilic nature of SNF coating (Supplementary Figs. 10, 11) and highly wetting surfaces of SNF-coated substrates (Supplementary Fig. 12) may create favorable membrane formation conditions, leading to enhanced quality of the generated PA rejection layer.

We further investigated the cross-sectional structure of the SNF20-NF0.5 membrane using scanning transmission electron microscope – energy dispersive X-ray (STEM-EDX). Figure 1E-1 presents the elemental mapping of fluoride (F) and nitrogen (N), in which F is the characteristic element in PVDF (C$_2$H$_2$F$_2$)$_n$, and N is the characteristic element for the SNF coating layer (amino acid in SNF) and PA rejection layer (−NH-C = O− group). To distinguish the SNF and PA layers, we further analyzed the cross-sections of the SNF0-NF0.5 membrane without SNF coating, which shows a PA layer spanning over the MF surface pore (Supplementary Fig. 13). Therefore, the N-riched region of the SNF20-NF0.5 membrane could reflect the successful deposition of SNF coatings attached to the substrate and the formation of PA rejection spanning over substrate pores. Furthermore, the enlarged bright-field TEM image (Fig. 1E-2) of the SNF20-NF0.5 membrane corroborates a continuous nanofilm with a thickness of merely 21 nm.

### Physicochemical properties of membranes
Figure 2A shows the FTIR spectra of PVDF substrates with/without SNF, and a series of SNF$_X$-NF membranes (with x indicating the amount of SNF loading). Compared with the pristine PVDF substrate, a characteristic peak of 1624 cm$^{-1}$ appeared in the SNF20-PVDF, which can be assigned to the Amide I band of SNF[28]. In addition, XPS elemental composition analysis with nitrogen elements appearing in

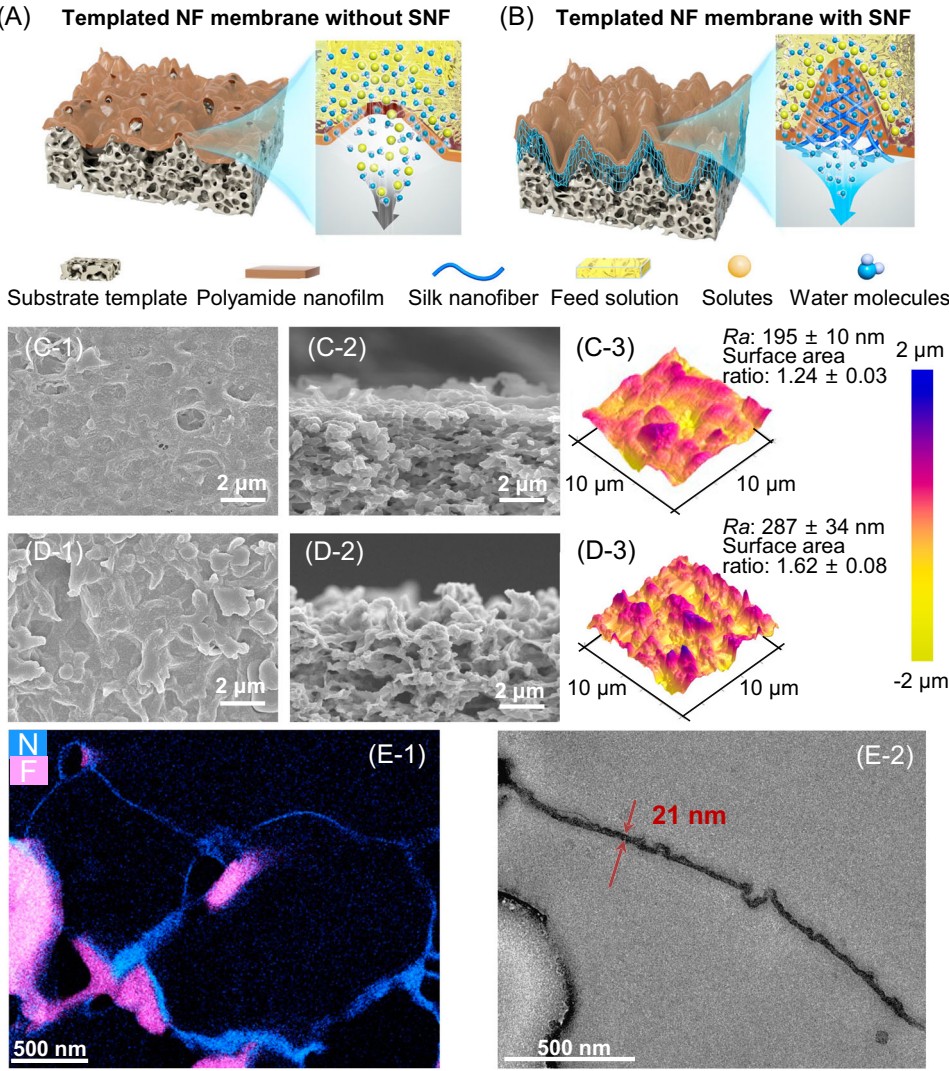

**Fig. 1 | The role of SNF in substrate-templated NF membranes.** Schematic diagrams of the water transport across the templated NF membranes without SNF (**A**) and with SNF (**B**). In both cases, substrate-templated crumple PA film formed on the substrate. SNF coating layer enhanced the integrity and roughness of PA films, thus improving the NF membrane separation performance. **C, D** Morphology changes of the substrate-templated NF membranes without SNF (SNF0-NF0.5) and with SNF (SNF20-NF0.5) formed by IP reaction of 0.5 wt% PIP with 0.1 wt% TMC. The SNF-coated PVDF substrates were fabricated by spraying 20 mL of SNF water suspension on the surface of pristine PVDF substrates, equivalent to a loading mass of 41 μg cm⁻². Surface (**C-1**) and cross-section (**C-2**) SEM image and AFM results (**C-3**) of SNF0-NF0.5. Surface (**D-1**) and cross-section (**D-2**) SEM images and AFM results (**D-3**) of SNF20-NF0.5. (**E-1**) STEM − EDX elemental mappings of SNF20-NF0.5, where "N" (bright blue) denotes nitrogen and "F" (pink) denotes fluorine. (**E-2**) Bright-field TEM images of the cross-section of SNF20-NF0.5.

SNF-modified substrates (Supplementary Table 1) further corroborated the FTIR results, confirming the successful loading of an SNF coating layer. After the IP reaction, the SNF0-NF0.5 membrane (without SNF layer) also had a weak peak at 1624 cm⁻¹, which can be attributed to the Amide I band for PA (piperazine amide)[34,39]. Interestingly, with the increased SNF loading, the resulting SNFₓ-NF membranes showed an intensified peak of the Amide I band, which can be attributed to amide groups in both SNF and rejection layers[34].

To further probe the properties of all NF membranes, we performed XPS characterization. Typically, the O/N ratio of a PA nanofilm ranges between 1 and 2, reflecting cross-linking degrees of the PA network from high to low. Compared with the control SNF0-NF0.5 membrane of an unusual O/N ratio (>2), SNF-NF membranes with SNF showed a typical O/N ratio in the range of 1–1.2 (Fig. 2B), similar to other TFC NF membranes reported in the literature[40]. The unusual O/N ratio of SFN0-NF0.5 could result from the extra oxygen contents. The additional oxygen contents could be possibly attributed to two sources. First, the hydrolysis of TMC in the IP reaction and the formation of PA oligomers could produce additional carboxyl groups in the PA network[41]. In addition, the oxygen-riched substrates (Supplementary Table 1, XPS result) could be detected by XPS through the defect regions of PA (Supplementary Fig. 7). To further clarify the influence of SNF on the properties of the PA, we employed Doppler Broadening Energy Spectroscopy (DBES) to compare free volume (or the size of sub-nanometer pores) in the PA layers of SNF0-NF0.5 (without SNF) and SNF20-NF0.5 (with SNF) membranes. Typically, a smaller S parament of DBES indicates a lower free volume. As shown in Supplementary Fig. 16, SNF20-NF0.5 possessed a smaller S value than SNF0-NF0.5, revealing the formation of a denser PA layer in the presence of the SNF coating. In addition, we investigated the surface zeta potential of SNF0-NF0.5 and SNF20-NF0.5 membranes (Fig. 2C). The results demonstrated that both membranes have a negatively charged membrane surface over a wide pH range, which can contribute to the rejection of charged solutes based on the Donnan effect.

### Role of SNF in the IP process
To illustrate the role of the SNF coating layer in the IP process, we further performed QCM-D tests to investigate its influence on the

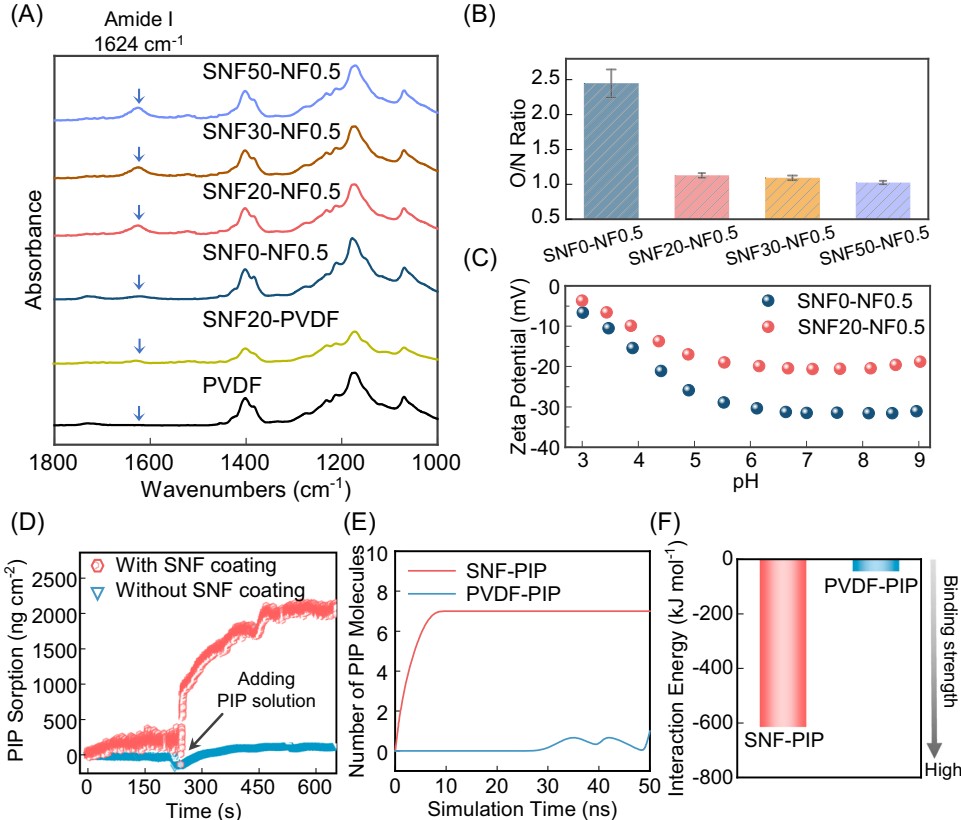

**Fig. 2 | Membrane physicochemical properties and molecular dynamics (MD) simulation results. A** FTIR spectra of pristine PVDF, SNF-coated PVDF substrates, and various SNF$_X$-NF membranes (with x indicating the volumetric loading of 129 μg mL$^{-1}$ SNF). **B** Comparison of the oxygen/nitrogen (O/N) ratio among nanofiltration membranes fabricated with increased SNF loading. The error bars represent the standard deviation of data from three distinct samples ($n = 3$). **C** Zeta potential of SNF0-NF0.5 and SNF20-NF0.5. **D** Quartz crystal microbalance with dissipation (QCM-D) sorption test of PIP monomer for bare and SNF-coated sensor. **E** The number of PIP molecules absorbed by PVDF and SNF and (**F**) The total interaction energy of SNF-PIP and PVDF-PIP based on MD simulation results at 50 ns. In both (**E**) and (**F**), sixteen PIP molecules were introduced and the MD simulations were performed over 50 ns.

sorption behavior of PIP monomers (Fig. 2D). Specifically, a quartz crystal sensor with or without SNF coating was mounted into the QCM-D chamber and rinsed with DI water. After stabilization, a PIP solution was introduced into the chamber. The change in frequency was then recorded to determine the PIP sorption. The PIP sorption by the SNF-coated sensor was nearly an order of magnitude higher compared to that of the control. To deepen the understanding of underlying mechanisms, we performed molecular dynamics (MD) simulations to explore the interaction between PIP and SNF at the molecular level. Figure 2E confirms the greater tendency of PIP sorption by SNF, which is underpinned by the more negative interaction energy in the SNF-PIP system (indicating a greater SNF-PIP intermolecular binding strength than PVDF-PIP as shown in Fig. 2F). Specifically, the prominent intermolecular force can be divided into three types: electrostatic interaction, hydrogen bonding, and van der Waals interactions[42]. Among these interactions, electrostatic interaction dominates the absorption process in the SNF-PIP system (Supplementary Fig. 18). In addition, the average number of hydrogen bonds in the SNF-PIP system is approximately six, which is significantly greater than the PVDF-PIP system with only one instantaneous hydrogen bond formed.

These preceding results demonstrate that the SNF coating layer can enhance the loading of PIP monomers onto the template substrate, which in turn enhances the IP reaction to form a better-quality PA nanofilm. In the IP process, the SNF coating layer can improve the storage of PIP, where PIP molecules can easily accumulate around the SNF. When contacting with the TMC-hexane organic phase during the IP reaction, the PIP-water aqueous layer can define the IP reaction interface layer to grow a crumple yet defect-free PA nanofilm along the

SNF coating layer, which is in good agreement with SEM images in Fig. 1D and XPS results in Fig. 2B. In addition to enhancing PIP loading, the SNF layer with an optimized thickness could preserve the rough topology of MF supports (Supplementary Fig. 6 and 9) to form a crumpled PA layer. Simultaneously, the interconnected porous structure of the SNF layer could further reduce tortuosity on water transport, resulting in significantly reduced hydraulic resistance. These combined effects contributed to the enhanced separation performance of SNF-NF membranes[37].

## Separation performance of NF membranes

Figure 3A shows the separation performance of SNF-NF membranes. After introducing the SNF coating layer, the Na$_2$SO$_4$ rejection significantly increased from <50% for control SNF0-NF0.5 membranes to 99.3 ± 0.1% for the SNF20-NF0.5 membranes. The enhanced Na$_2$SO$_4$ rejection of the SNF20-NF0.5 membrane resulted from the SNF coating layer that improves IP reaction conditions (QCM-D results in Fig. 2D) to form a defect-free rejection layer (SEM examination in Fig. 1C/D-1). Nevertheless, a further increase in SNF loading reduced the water permeance in SNF-NF membranes (Supplementary Fig. 21A). This could be attributed to an enhanced tortuosity in thicker SNF layer (Supplementary Fig. 6 and Supplementary Fig. 9) with increased hydraulic resistance of substrates (Supplementary Fig. 22), highlighting the importance of optimizing SNF loading mass.

We further optimized membrane separation performances by manipulating the IP conditions[43], e.g. reducing reaction time from 60 to 30 s and decreasing PIP concentration from 0.5 to 0.2, 0.1 wt%. Notably, the SNF20-NF0.1* membrane with a thinner PA of

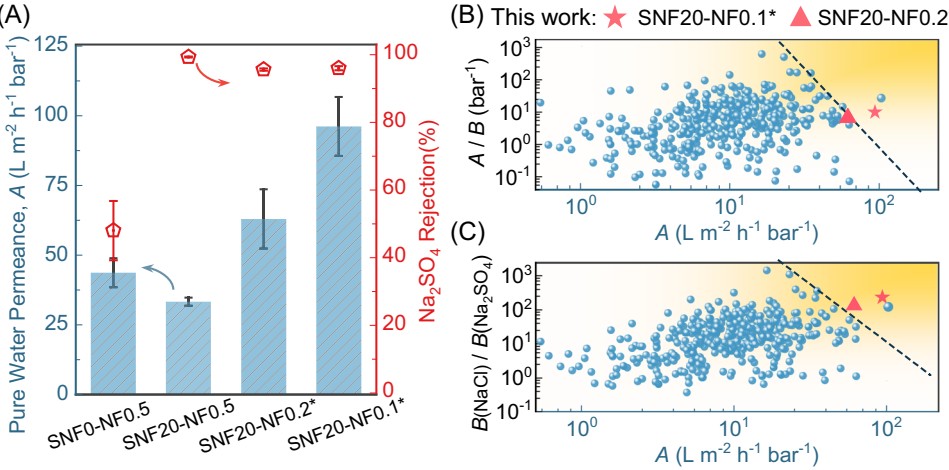

**Fig. 3 | Desalination performance of SNF-NF membranes under pressure-driven cross-flow filtration mode. A** The influence of SNF coating and IP condition on pure water permeance and $Na_2SO_4$ rejection. The error bars of separation performance represent the standard deviation of data from three distinct samples ($n = 3$). The SNF0-NF0.5 without SNF coating and SNF20-NF0.5 with SNF coating (20 mL of SNF suspension) kept the same IP condition: 0.1 wt% of TMC and 0.5 wt% of PIP; reaction time of the 60 s. To further optimize membrane separation performance, the PIP concentration for fabricating SNF20-NF0.1* and SNF-NF0.2* membranes was reduced to 0.2 wt% and 0.1 wt%, respectively, and IP reaction time was reduced

to 30 s. The rejection test was performed using a feed solution of 1000 ppm $Na_2SO_4$ and pure water permeance was determined using DI water. The filtration tests adopted a conventional pressure-driven cross-flow mode with an applied hydraulic pressure of 3 bar. **B** The trade-off relationship between the water permeance and water-salt ($A/B_{Na2SO4}$) selectivity of SNF0-NF0.1* and SNF20-NF0.2* membranes compared to other literature data[44]. **C** The trade-off relationship between the water permeance and NaCl/$Na_2SO_4$ selectivity of SNF20-NF0.1* and SNF20-NF0.2* membranes compared to literature data[44].

approximately 14 nm (Supplementary Fig. 23) demonstrates excellent water permeance of $96.2 \pm 10 \, L \, m^{-2} \, h^{-1} \, bar^{-1}$ with a $Na_2SO_4$ rejection of $96.0 \pm 0.6\%$ (Fig. 3A). Such ultra-high water permeance, approximately one order of magnitude higher than the existing commercial NF membranes[6], can be ascribed to the highly porous and rough SNF-coated MF substrates. Specifically, the SNF layer in NF membranes can reduce the water transport resistance by mitigating the funnel effects and enlarge the effective filtration area by facilitating the formation of a crumpled PA rejection layer (Supplementary Fig. 25). To further prove the effectiveness of the crumpled structure, we coated the SNF layer on a smoother MF substrate (-$R_a$ of 90 nm, Supplementary Fig. 26) with higher water permeance of $19360 \pm 1550 \, L \, m^{-2} \, h^{-1} \, bar^{-1}$ (Supplementary Fig. 27). The resulting NF membrane had a much lower water permeance on the order of $15.0 \, L \, m^{-2} \, h^{-1} \, bar^{-1}$ (Supplementary Fig. 28).This experimental observation underpins the importance of a crumpled rejection layer in improving water permeance[36].

The separation properties of NF membranes are typically constrained by a permeance-selectivity trade-off, with the lines in Fig. 3B−C representing the upper bounds for permeance-water/$Na_2SO_4$ selectivity and permeance-NaCl/$Na_2SO_4$ selectivity, respectively. In our work, the ultra-permeable SNF20-NF0.1* membrane exhibits excellent separation performance that outperforms conventional PA NF membranes by a large margin. In the membrane community, breaking the upper bound could potentially translate into a development of membrane applications, such as pretreatment in seawater desalination, groundwater treatment, etc.[21,44]. In this study, we further explore the use of the ultra-permeable NF membrane for a submerged vacuum-driven NF process.

**Submerged vacuum-driven NF process**

Figure 4A illustrates a submerged vacuum-driven NF process. An aeration device releases air bubbles to mitigate concentration polarization and clean the membrane surface. In this vacuum-driven process, the feed solution was sucked through the NF membrane to produce purified permeate. To validate SNF-NF membranes in the submerged vacuum-driven system, we examined the membrane separation performance. It is worthwhile to note, for vacuum-driven nanofiltration, its maximum driving force is limited (<1 bar). Therefore, this process is more suitable to feed solutions with relatively low

osmotic pressure. We chose a $Na_2SO_4$ concentration of 500 ppm in the vacuum-driven process, as the osmotic pressure is representative of that for municipal wastewater treatment[13]. As shown in Fig. 4B, the SNF20-NF0.1* membrane exhibited a water flux of $56.8 \pm 7.1 \, L \, m^{-2} \, h^{-1}$ at a suction pressure of 0.9 bar. In comparison, the flux of a commercial membrane NF270 was ten times lower ($4.8 \pm 0.2 \, L \, m^{-2} \, h^{-1}$), making it not viable for many practical applications (e.g., submerged MBR). On the other hand, even though the SNF0-NF0.1* membrane without SNF had higher flux compared to the SNF20-NF0.1* membrane, the former exhibited a low $Na_2SO_4$ rejection of merely 20%. Without the SNF coating, the high tendency of forming defects in the PA layer (Supplementary Fig. 7) makes this membrane less suitable for NF applications.

Figure 4C demonstrates SNF20-NF0.1* membrane simultaneously maintained a $Na_2SO_4$ rejection of > 96% and a high passage of essential minerals $Ca^{2+}$ and $Mg^{2+}$, resulting in improved minerals-sulfate selectivity compared to NF270 and SNF0-NF0.1* (Supplementary Fig. 30). A high selectivity of divalent cations to sulfate is important for minimizing the risk of membrane scaling[4]. In addition, $Ca^{2+}$ and $Mg^{2+}$ are essential minerals for human body[45]. In addition to minerals, polyfluoroalkyl substances (PFASs) have drawn critical environmental concerns, which are toxic and bioaccumulative compounds even at trace concentrations (ppb), thereby posing major health risks to the public[46]. Strikingly, our SNF20-NF0.1* membrane also achieved high rejection against a series of PFASs, e.g., PFOS rejection of $99.6 \pm 0.2\%$ (Fig. 4C and Supplementary Fig. 31), implying its great potential for treating PFAS-contaminated water.

Ultra-permeable NF membranes are expected to significantly lower the SEC and GHG emissions, particularly for treating less saline water/wastewater[15,18]. We further analyzed the SEC of the vacuum-driven system for membranes with different permeances (Fig. 4D). Compared to the commercial benchmark (NF270), the ultra-permeable SNF20-NF0.1* membranes offer approximately 80% reduction in the SEC. Figure 4D also illustrates the required water permeance for enabling the vacuum-driven operation. Assuming a target water flux of $25 \, L \, m^{-2} \, h^{-1}$ and a maximum vacuum of 1 bar, the minimum required permeance is approximately $30 \, L \, m^{-2} \, h^{-1} \, bar^{-1}$. In practice, the operational vacuum is often much lower, e.g., due to air

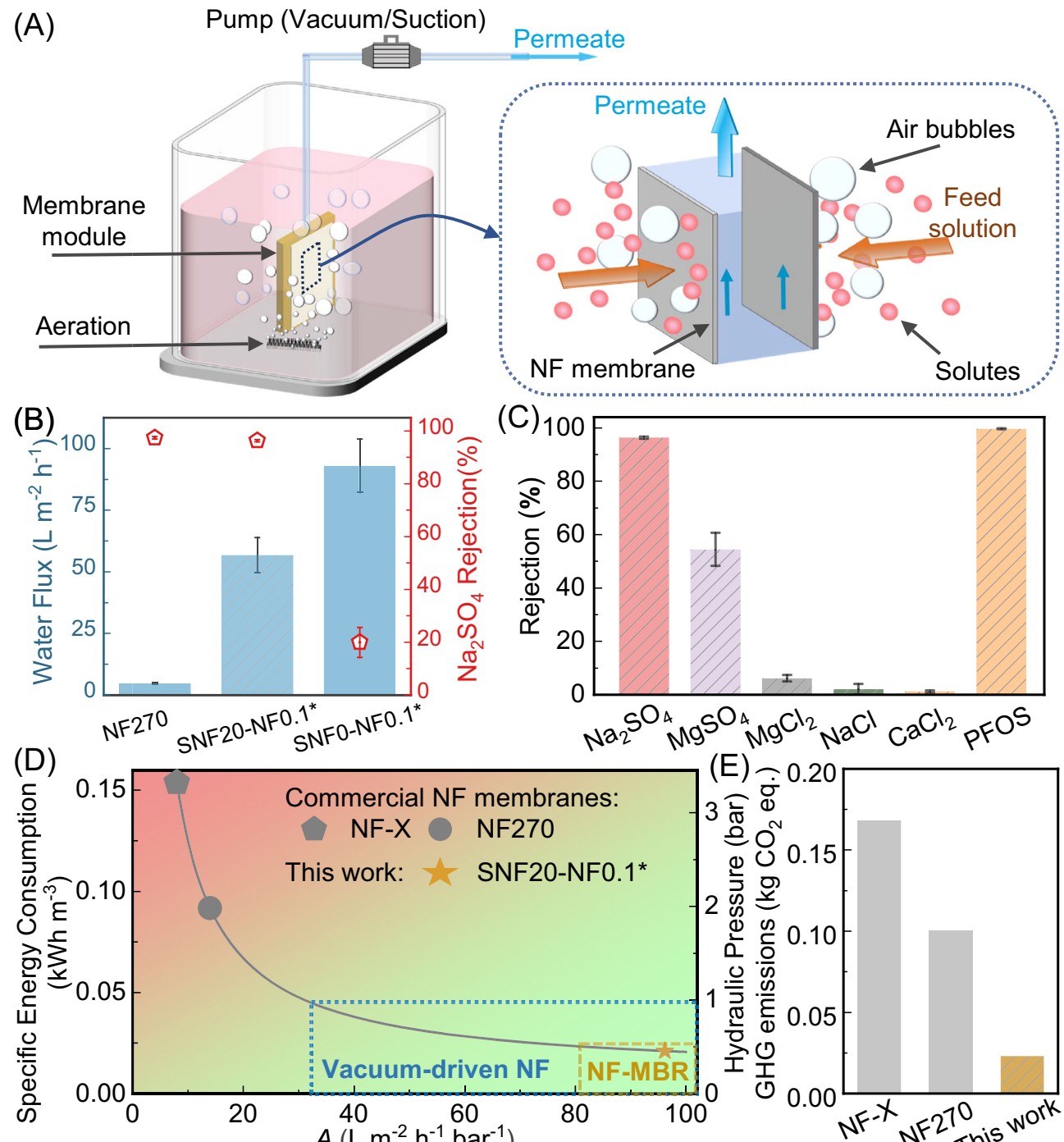

**Fig. 4 | Separation performance of NF membranes under a vacuum-driven filtration mode. A** The schematic illustration of a submerged vacuum-driven nanofiltration process and (**B**) the comparison of separation performance among commercial NF270 membranes, SNF20-NF0.1* and SNF0-NF0.1* under vacuum filtration using a feed solution of 500 ppm $Na_2SO_4$. **C** All filtration tests were performed under vacuum-driven filtration mode with an applied vacuum pressure of 0.9 bar. For salt rejection, the feed solution contained a single salt type of 500 ppm. For PFOS, the feed solution contained 200 ppb PFOS in a 10 mM NaCl background solution. The error bars of separation performance represent the standard deviation of test data from three distinct samples ($n = 3$). **D** The relationship between water permeance ($A$, L m$^{-2}$ h$^{-1}$ bar$^{-1}$), the specific energy consumption (SEC, kWh m$^{-3}$), and applied hydraulic pressure (bar). The calculation of SEC is based on our previous studies[18,44], assuming a transmembrane osmotic pressure difference of 0.1 bar, a water recovery of 60%, and a target water flux of 25 L m$^{-2}$ h$^{-1}$. The analysis did not include energy recovery devices (ERD), since ERD is rarely implemented in NF processes. (**E**) Comparing the greenhouse gas (GHG) emissions of NF processes adopting commercial membranes (NF-X and NF270) and the NF membrane prepared in this work (SNF20-NF0.1*), respectively.

leakage or in order to avoid high SEC. In the context of MBR, the vacuum pressure if generally lower than 0.5 bar[13], calling for water permeance of > 80 L m$^{-1}$ h$^{-1}$ bar$^{-1}$—a value far beyond those of existing commercially available membranes. In addition to greatly reduced SEC, the vacuum-driven NF system confers several noteworthy technical and economic benefits, including eliminating additional equipment costs (without the need for pressure vessels), eradicating spacer-induced membrane fouling, reducing footprint, and allowing easy retrofit to existing treatment processes. Furthermore, Fig. 4E reveals a potential 80% reduction in GHG emissions for the ultra-

permeable NF membranes compared to the commercial benchmark NF270. As a result, ultra-permeable NF membranes offer huge potential for achieving more energy-efficient and greener water treatment.

In this work, we fabricated an ultra-permeable yet defect-free SNF-NF membrane and applied it in a submerged vacuum-driven filtration system for water purification. Strikingly, the SNF-NF membrane exhibited an ultra-high water permeance of 96.2 L m$^{-2}$ h$^{-1}$ bar$^{-1}$ together with excellent water/solute and solute/solute selectivity. The outstanding membrane separation performance stems from the crumple and defect-free PA nanofilm formed on the SNF-coated MF substrate which favors the PA formation and overcomes the funnel effect of conventional TFC membranes. Furthermore, we designed a submerged vacuum-driven filtration system adopting ultra-permeable SNF-NF membranes, which simultaneously achieved high water flux, excellent rejection against salts and contaminants, and significantly reduced SEC and GHG emissions. These intriguing features could potentially revolutionize the current MBR system in water/wastewater treatment plants by retrofitting the MF/UF membranes with our SNF-NF membranes (NF-MBR).

It is worthwhile to note that, for conventional interfacial polymerization, decreasing PIP concentration is often adopted in order to enhance water flux. Nevertheless, this primitive strategy greatly increases risks of forming more defects in the PA rejection layer[47–49]. The tendency of defect formation is largely due to insufficient supply of PIP monomers[50]. In contrast, the SNF layer offers major advantages over the conventional approach. SNF increases the sorption of PIP, which allows it to act as a "PIP reservoir" to ensure sufficient supply of amine monomer (Fig. 2D). At the same time, the SNF layer provides a more defined reaction interface with slower release of the PIP monomer (Supplementary Fig. 20). Due to these combined effects, the SNF achieves simultaneously moderate effective PIP concentration at the IP reaction interface (which is favorable for forming a more permeable NF membrane) and sufficient PIP storage (which minimizes defect formation)[51]. In addition, the large PIP storage in SNF, together with the reduced surface pore size of SNF-coated substrates, allows the adoption of lower PIP concentrations for IP reaction while still maintaining high rejection of Na$_2$SO$_4$ (Fig. 3A).

## Methods

### Preparation of SNF

The preparation of SNF can refer to the reported protocol[33]. Specifically, as shown in Supplementary Fig. 1, the fabrication process of SNF mainly involves degumming of the raw silk, hydrolysis of the degummed silk fibers, and ultrasonic breaking of hydrolyzed silk. To obtain the degummed silk fibers, the raw silkworm silk was boiled in a 0.5 wt% Na$_2$CO$_3$ solution to remove sericin proteins completely. Then the degummed silk (1 g) was cut into short fiber (-length of 1 cm) and mixed with H$_2$SO$_4$ solution (40 wt%, 50 mL). The mixture was heated at 60 °C for 2 h with a continuous stirring to hydrolyze the degummed silk. To further break the silk fibers, the mixture was mechanically disintegrated by an ultrasonic homogenizer for 20 min (JY-98 IIIN, Ningbo Xingzhi Technology Co., Ltd, China). The resulting SNF suspension was then centrifuged at 9390 g for 10 min to remove the unfibrillated fraction and the pH of SNF suspension was adjusted to 10 and the suspension was stored at 4 °C before further use.

### Preparation of SNF-coated substrates

To uniformly coat SNF coating on the substrates, a spray coating technique was adopted using an N$_2$ spray gun (RL-90-F/R, Prona Industries Co., Ltd. China) on a vacuum plate (946A-3, Guangzhou YIHUA Co., Ltd, China). To optimize the SNF loading, 20 mL, 30 mL, and 50 mL of SNF suspension (129 µg mL$^{-1}$) were spray-coated on the surface of a hydrophilic PVDF MF substrate (63.5 cm$^2$) with care, resulting in mass loadings of 41 µg cm$^{-2}$, 61 µg cm$^{-2}$ and 102 µg cm$^{-2}$, respectively. The SNF-coated substrates were then dried at 45 °C for

10 min in an oven. More details regarding the loading mass of each substrate can be found in Supplementary Table 2.

### Preparation of SNF-NF membranes

To fabricate the NF membranes, the interfacial polymerization method was utilized to form a PA layer on the SNF-coated substrate. Specifically, the PA layer was formed at the interface between the PIP/water solution (with different concentrations: 0.1 wt%, 0.2 wt%, and 0.5 wt%) and TMC/hexane (0.1 wt%) solution. To initiate the IP reaction, an SNF-coated substrate was first immersed in the PIP solution for 60 s. After removing the excess PIP solution with a rubber roller, the PIP-soaked substrate was immersed in the TMC/hexane solution for a specific time (15, 30, or 60 s) to grow a PA layer without further post-treatment. The resulting SNF-NF membranes were stored in water at 4 °C for further characterization. More details regarding the IP conditions and the corresponding sample names can be found in Supplementary Table 3. For example, SNF20-NF0.5 means the NF membranes fabricated on a substrate (63.5 cm$^2$) coated with a volume of 20 mL SNF (129 µg mL$^{-1}$) using a 0.5 wt% PIP solution in IP reaction

### Characterization

Scanning electron microscopy (SEM, S4800, Hitachi, Japan) was used to characterize the surface and cross-sectional morphology of the membranes. Before the SEM characterization, samples were sputter-coated with a thin gold layer via a sputter coater (SCD 005, BAL-TEC). The cross-sectional structure of the NF membranes and the surface morphology of SNF were assessed by transmission electron microscopy (TEM, Philips CM100, Eindhoven, Netherlands). STEM-EDX (Thermo Scientific Talos F200X) was used to investigate the elemental distribution in the cross-section structure of NF membranes. The surface roughness of the membranes and the dimension of SNF were measured by atomic force microscopy (AFM, Multimode 8, Bruker, MA), and the data were processed by using analytics software (NanoScope Analysis). The surface charge of membranes over a pH range from 3 to 10 was characterized by a streaming potential analyzer (SurPASS 3, Anton Paar). Fourier transform infrared spectroscopy (FTIR, Nicolet 6700 FTIR spectroscopy, Thermo Fisher Scientific, Waltham, MA) was applied to determine the surface functional groups of SNF and membranes. A water contact angle analyzer (Attention Theta, Biolin Scientific, Sweden) was used to measure the dynamic water contact angle of the membranes. X-ray photoelectron spectroscopy (XPS, Kratos AXIS Ultra DLD) was applied to examine the elemental content of the membranes.

To probe the sorption capability of the SNF coating layer, a quartz crystal microbalance with dissipation (QCM-D, E4, Q-Sense Biolin Scientific, Sweden) was used. Before the QCM-D test, we prepared an SNF-coated gold sensor by spraying the SNF suspension at an optimized mass loading at 41 µg cm$^{-2}$ based on membrane separation performance tests (Fig. 3A). Then, the SNF-coated sensor was immersed in methanol for stabilizing and completely dried in the oven at 45 °C. The obtained SNF-coated sensor was directly mounted in a QCM-D chamber and stabilized in DI water. After stabilization, the PIP/water solution (0.5 wt%) was injected into the chamber, waiting for the sensor to be stable again. The QCM-D records the frequency change of the sensor throughout the entire experiment, which can be transformed into the deposed mass change of PIP by Q-Tool software (Q-Sense, Biolin Scientific, Sweden). For comparison, the sorption test for a control sensor (without SNF coating) was also conducted under the same experimental conditions.

The MWCO of NF membranes was determined by filtration test of a series of polyethylene glycol (PEG) molecules (molecular weight: 400, 600, 800, and 1000). Four PEG aqueous solutions with a single concentration of 200 ppm were used as the feed solution to be filtrated by NF membranes, separately. The PEG concentrations in feed and permeate were detected by a TOC analyzer (TOC-V CPH,

Shimadzu), which can obtain the rejection for PEG molecular. Based on the rejection cure of NF membranes for a series of PEG solutions, MWCO was according to the molecular weight at the rejection of 90%.

## Molecular simulations

To explore the molecular interaction between PIP and SNF, we performed MD simulations of the absorption processes by using Gromacs 2019.4 package[52]. General amber force field parameters[53] and RESP (restrained electrostatic potential atomic partial) charges[54] were used and generated by the ANTECHAMBER program in AmberTools for an SNF protein fragment (3UA0) and PIP moleculars[55]. The TIP3P water model was used for water molecules. At the beginning of the simulation, one SNF protein fragment was placed in a simulation box of 8.17 × 8.17 × 8.17 nm³. Then, seventeen protonated PIP molecules were randomly inserted into the box. Subsequently, 17067 water molecules were filled into the simulated box to simulate the water environment. Finally, NPT (constant-pressure, constant-temperature) runs were performed at 303 K for 50 ns. In contrast, 386 repeating units of PVDF fragment, having similar molecular weight to the SNF protein fragment, was selected as a model molecule to replace protein molecules in the system. At the beginning of the simulation, a PVDF fragment was placed in the simulation box of 10.11 × 10.11 × 10.11 nm³. seventeen protonated PIP molecules were randomly inserted into the box. Simultaneously, seventeen chloride ions were added to the system as counterbalance ions. Then, 32759 water molecules were filled into the simulated box to simulate the water environment. Finally, NPT runs were performed at 303 K for 50 ns. Visualization of the molecular structures is made by using Visual Molecular Dynamics software.

## Separation performance in the conventional cross-flow filtration system

Membrane separation performances were examined using a homemade cross-flow filtration setup at the cross-flow velocity of 22 cm s⁻¹ and the temperature was kept at 25 ± 0.5 °C. The effective filtration area of each membrane cell was 2 cm². All NF membranes were pre-stabilized at a hydraulic pressure of 3 bar for 30 min. Membrane water permeance of membranes was calculated by the Eqs. (1) and (2)

$$J_w = \frac{\Delta V}{\Delta t \times S} \tag{1}$$

$$A = \frac{J_w}{\Delta P - \Delta \pi} \tag{2}$$

where $J_w$ (L m⁻² h⁻¹) was the water flux, $\Delta t$ (h) was the filtration time, $S$ (m²) was the effective membrane area, $\Delta V$ (L) was the permeate volume. $A$ (L m⁻² h⁻¹ bar⁻¹) was the membrane water permeability constant. $\Delta P$ was the transmembrane osmotic pressure and $\Delta \pi$ (bar) was the transmembrane osmotic pressure. The salt rejection was tested using a salt feed solution of 1000 mg L⁻¹ and can be calculated by Eq. (3)

$$R = \frac{C_f - C_p}{C_f} \tag{3}$$

where $C_f$ and $C_p$ were the concentration of salts in the feed and permeated solution, respectively. A conductivity meter (Ultrameter II, Myron L Company, Carlsbad, CA) was for determining the salt concentration.

## Separation performance of NF membranes in the submerged vacuum-driven nanofiltration system

The schematic of a submerged vacuum-driven membrane filtration system is shown in Fig. 4A, where a peristaltic pump (BT1003J, Longer Pump, China) provides a transmembrane pressure at 0.9 bar to drive the nanofiltration process. The temperature of feed solutions was

maintained at 25 ± 0.5 °C. To mitigate the membrane concentration polarization in the vacuum filtration test, an aeration system was employed to continuously generate air bubbles, effectively flushing the surface of the NF membrane. The off-gas volume for this aeration system was maintained at a rate of 0.5 L min⁻¹. The effective filtration area of the NF membrane is 8 cm², which was used in all submerged vacuum-drive nanofiltration experiments.

To assess the NF separation performance under the submerged vacuum-driven membrane configuration, we measured the membrane rejection of various salt (Na₂SO₄, MgSO₄, MgCl₂, NaCl, CaCl₂; single salt centration of 500 ppm) and organic micropollutants (PFASs) respectively. The rejection of PFASs was tested by a mixed feed solution containing PFOS, PFBA, PFBS, GenX, and PFOA at a single concentration of 200 ppb and 10 mM NaCl solution as a background solution. The PFASs concentration was determined by using liquid chromatography with tandem mass spectrometry (LC-MS/MS, 1290 Infinity, Agilent; 3200 QTRAP, AB SCIEX, Singapore)[56].

## Calculation of specific energy consumption and greenhouse gas emissions

The specific energy consumption (SEC) for water purification is defined as the amount of energy required to produce one cubic meter of purified water. In this study, we obtained the SEC values using the method adopted from the reference[18]. The specifical input parameters were described in Supplementary Table 4. In addition, the life cycle impact assessment of electricity consumption was conducted using the LCA software GaBi (version 7). The greenhouse gas emissions were calculated as a midpoint result, following the Centre for Environmental Studies (CML) classification method (CML2001 v. Jan. 2016)[57]. This approach allows for a comprehensive evaluation of the environmental impact associated with electricity consumption throughout its life cycle.

## Data availability

All data generated in this study are provided in the article, Supplementary Information, and Source Data file. All data are available from the corresponding author upon request. Source data are provided with this paper.

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

## Acknowledgements

The work was substantially supported by the Innovation and Technology Fund of the Hong Kong Special Administration Region, China (GHP/181/20GD and ITS/249/20). Partial support was also received from the Research Grants Council of the Hong Kong Special Administration Region, China (GRF 17201921 and SRFS2021-7S04). Z.Y. is also supported by the ARC Discovery Early Career Researcher Award (DE230100114) from the Australian Research Council in Australia. The QCM-D work was supported by The University of Hong Kong (URC Small Equipment Fund 102010138 and Seed Funding for Strategies Interdisciplinary Research Scheme 102010174).

## Author contributions

B.G., Z.Y., and C.T. conceived the idea and designed the research. B.G., L.Z., W.L., L.L., and L.W. performed experiments. H.G. and X.S. provided constructive suggestions for results. B.G., L.P., L.Z., Z.Y., and C.T. contributed to writing the manuscript.

## Competing interests

The authors declare no competing interest.
