## [Peer Review File · Nature Communications]

Ultra-permeable Silk-based Polymeric Membranes for Vacuum-driven NanofiltrationReviewers' Comments:

Reviewer #1:

Remarks to the Author:

This manuscript reports a high-performance vacuum-driven NF process with the installation of the ultra-permeable NF that cannot be achieved by current commercial membranes. In addition to the new materials designed in this work, the novelty of this work lies in the way that the membrane filtration was driven by down-stream vacuum rather than a positive pressure on the upstream side of the membrane. The manuscript is well-written and I read the manuscript with interest as it offers a new energy-efficient and sustainable water treatment process and may contribute to the broader scientific community. Hence, the manuscript can be considered in Nature Communications after a revision, as some issues require further clarifications. The detailed comments are listed below:

1. The method section described the fabrication methods: "To obtain the degummed silk fibers, the raw silkworm silk was boiled in a 0.5 wt% Na₂CO₃ solution to remove sericin proteins completely..." What is the composition of raw silk? I suggest that the authors should provide a clear description of the structure of silk so that readers can understanding more information of silk nanofibers.
2. In the manuscript, it is mentioned that "the SNF coating mass can alter the surface morphology of the substrate". What were the morphology changes of SNF-NF membranes with the additional SNF? More detailed characterization results should be provided.
3. Figure 2F shows a greater SNF-PIP intermolecular binding strength than PVDF-PIP. I suggest showing the relationship between interaction energy and binding strength more clearly and directly in Figure 2F.
4. In Fig. 1D, the later membrane demonstrated a higher surface area ratio. However, the control SNF0-NF0.5 exhibited higher water permeance than the SNF20-NF0.5 membranes. Please provide more convincing evidence and explanations for this experimental result.
5. Figure 4A illustrates the submerged vacuum-driven nanofiltration process. However, the construction of the filtration equipment is not entirely clear. I suggest the authors provide a digital photograph of the vacuum filtration apparatus to enhance understanding. Additionally, please clarify the installation of the NF membranes and methods for controlling vacuum pressure.
6. The authors changed the Na₂SO₄ concentration to 500 ppm in the vacuum-driven NF filtration test. Please explain why choosing such specific concentration which differs from many other studies using 1000 ppm concentration for their testing.
7. Figure 4C presents the separation performance of SNF20-NF0.1* and NF270 for five salts using a submerged vacuum-driven filtration process. What's the separation performance of these membranes in conventional crossflow filtration mode?
8. The authors have calculated the specific energy consumption (SEC) based on previous studies; however, some assumptions have changed. To ensure clarity and enable readers to better understanding the calculation process, the authors should describe the method used for calculating SEC in this study.
9. In your results, the SNF20-NF0.1* membranes achieved a PFOS removal of 99.6%, which was considerably higher than the removal rates for other PFASs. Please explain this significant difference!
10. In Supplementary Figure 11, the water permeance is observed to decrease rapidly from SNF20-

PVDF to SNF50-PVDF. Please clarify the factors contributing to the decline in water permeance.

Reviewer #2:

Remarks to the Author:

This paper presents nanofiltration membranes with ultrafast water permeance by using a SNF gutter layer. The SNF layer preserves the rough topology of microfiltration supports while absorbs enough amount of amine molecules at low concentrations, thereby facilitating interfacial reaction for the creation of ultrathin nanofilms. The approach is novel, and the performance is outstanding among literature reported membranes. Therefore, I would suggest this paper to be considered for publication in Nature Communications, if the comments below can be addressed.

1. The abstract should be restructured. The key advance in this paper is not the vacuum-driven nanofiltration since it has not been proved as a practical alternative to the pressure-driven nanofiltration in industry. Rather, the authors should shift the focus to the use of SNF layer for enhancing the water permeance in nanofiltration, which is indeed the focus throughout the main manuscript.
2. The SNF layer should contribute to the membrane formation mechanism in three aspects: i) preservation of the rough topology of MF supports; ii) enhanced absorption of PIP solutions; and iii) interconnected porous structure with reduced tortuosity. However, the authors only attributed the performance difference to the surface roughness, which could be misleading. For example, SNF30-PVDF supports have similar surface roughness to SNF20-PVDF supports (Supplementary Fig. 4), but the permeance of coated membranes SNF30-NF0.5 is significantly lower than that of SNF20-NF0.5 (Fig. 3 and Supplementary Fig. 10). This could be attributed to an enhanced tortuosity in thicker SNF layer (please provide the thickness of SNF layer with various loading mass), rather than their difference in surface roughness.
3. The enhanced absorption of PIP solutions enabled the fabrication of ultrathin nanofilms at very low monomer concentrations (i.e., SNF20-NF0.2* and SNF20-NF0.1*), leading to ultrahigh water permeance (Fig. 3). The active thickness of polyamide layer in these membranes (i.e., SNF20-NF0.2* and SNF20-NF0.1*) should be measured to support this phenomenal increase in their permeances.
4. Following the question above, thicker but defect-free polyamide layer should be fabricated on the pristine PVDF (SNF0) support at higher PIP concentration. With the measured thickness of active layer, it can further prove the advance of SNF layer for the creation of ultrathin polyamide films.
5. Line 273-276, the comparison and the conclusion is not convincing. Apart from the difference in surface roughness, PES UF substrates should have a lower water permeance (please provide this data) than PVDF MF substrates, which could significantly contribute to the reduction of water permeance in the composite membranes.
6. The upper bound in Fig. 3 needs to be updated. For example, the results from Ref. 20 should be added to the plot.

Reviewer #3:

Remarks to the Author:

The authors tried to develop highly permeable nanofiltration (NF) membranes, which are available for vacuum-driven NF, by spray-coating silk nanofiber (SNF) on a PVDF MF membrane. However, the manuscript has many problems, as follows:

- a) some interpretations are contradictory
- b) some are misled into wrong interpretations
- c) some suffer from the lack of empirical evidence or appropriate/fair comparisons

Therefore, the reviewer cannot agree with the publication of this manuscript in Nature Communications.

The details on the above comments are as follows.

Major comments

(1) Page 5, Lines 89-95

The SNF-incorporated NF membrane exhibited 96.2 LMH/bar while maintaining Na₂SO₄ rejection of 96% under crossflow filtration, while it achieved water flux of 56.8 LMH and Na₂SO₄ rejection of 96.3% at a partial vacuum of 0.9 bar.

The reviewer raises an issue with the similar Na₂SO₄ rejection despite the significant difference in the water flux (i.e., 96.2 LMH/bar vs. 56.8 LMH at a partial vacuum of 0.9 bar).

In the two processes, the transport of water molecules would be governed by hydraulic pressure, whereas ion transport should be governed by ion concentration polarization.

As such, since water molecules and ions move by different driving forces, the significant difference in the water flux was likely to reduce apparent Na₂SO₄ rejection on the condition that the feed solutions' concentrations were the same.

The authors need to think over this issue.

(2) Fig. 1(c) and 1(d)

The authors need to provide the roughness data of the control PVDF MF membrane's surface.

(3) Page 7, Lines 119-123

According to conventional wisdom, organic materials smaller and larger than 0.45 μm are considered dissolved organic matter (DOM) and particulate matter, respectively. In this regard, SNF, which was 380 nm in length, corresponds to DOM, indicating that the authors spray-coated DOM on the MF membrane's surface without chemical crosslinking. Can the authors guarantee SNF is not leaked to a permeate solution during filtration?

(4) Page 8, Lines 145-149

The authors insisted that the hydrophilic nature of SNF may be beneficial for creating favorable membrane formation conditions, while the dynamic contact angle results demonstrated better water distribution ability of the SNF-coated substrate based on Supplementary Fig. 6. However, according to Supplementary Fig. 6, the contact angle values of the control PVDF and the SNF20-PVDF membrane were approaching 0. Accordingly, the contact angle data of the SNF20-PVDF membrane cannot be used to corroborate the hydrophilic nature of SNF. The two contact angle data were highly likely to be associated with highly wetting surfaces stemming from the highly porous

surface of the two MF membranes rather than their hydrophilicity.

(5) Page 10, Lines 189-205

The authors presumed that the crosslinking degrees of the SNF-NF membranes with SNF were higher than the SNF0-NF0.5 membrane based on their lower O/N ratio and lower negative surface zeta potential. However, the O/N ratio and the surface zeta potential cannot directly support the higher crosslinking of the SNF-NF membranes with SNF due to the following reasons. First, SNF also has a lot of nitrogen in its repeating unit. Thus, the authors cannot conclude that the SNF-NF membrane with SNF showed higher crosslinking degree without consideration of the nitrogen content contribution of SNF. Second, SNF is known to possess more OH groups than carboxyl groups, which is stark contrast to polyamide containing much more carboxy groups than hydroxyl groups. Accordingly, SNF itself can endow the SNF-NF membranes containing SNF with lower negative surface zeta potential regardless of crosslinking degree of polyamide. Lastly, the authors attributed the high O/N ratio of the SNF0-NF0.5 membrane to the extra oxygen content of the PVDF substrate. However, considering the polyamide film's thickness (i.e., ca. 20 nm) and the detection depth of XPS, the O/N ratio of the SNF0-NF0.5 is thought not to arise from the PVDF substrate's oxygen content.

(6) Fig. 3(a)

Based on Fig. 3(a), the reviewer thinks that the effects of reducing interfacial polymerization (IP) time and PIP content were much more dominant than that of SNF coating. The authors should provide data regarding the water permeance and Na₂SO₄ rejection of the SNF0-NF0.2* and SNF0-NF0.1* to clarify the dominant factor influencing the filtration performance.

(7) Page 11, Lines 215-234

The authors insisted that the SNF coating layer can enhance the loading of PIP monomers onto the template substrate, subsequently enhancing the IP reaction to form a better-quality PA nanofilm. However, Fig. 3(a) displayed that the lower content of PIP led to higher water permeance while maintaining comparable Na₂SO₄ rejection. The water permeance data shown in Fig. 3(a) looks contradictory to the above authors' claim, considering that the lower content of PIP should result in the lower loading of PIP onto the template substrate during IP. The authors should address this contradictory issue in their interpretation.

(8) Fig. 4(b) and 4(e)

The authors compared the water flux of the SNF20-NF0.1* only with the NF270, which should be prepared with a different type of substrate (e.g., a dense ultrafiltration membrane). Hence, the authors need to provide the water flux of the SNF0-NF0.1* for a fair comparison and further verification of the SNF coating effect. It is also the case for the comparison of the GHG emissions. In the comparison of the GHG emissions, the data regarding the SNF0-NF0.1* should be provided in parallel with the SNF20-NF0.1*.

Minor comments

1) Page 8, Line 142

It can be difficult for readers to understand the meaning of SNF20-NF0.5. Accordingly, it is

advisable to briefly explain the meaning of the sample names in the Methods.

2) The mean pore size of the PVDF MF membrane

The authors need to provide the mean pore size of the PVDF MF membrane.

**Response to the comments:**

**Reviewer #1 (Remarks to the Author):**

*This manuscript reports a high-performance vacuum-driven NF process with the installation*
*of the ultra-permeable NF that cannot be achieved by current commercial membranes. In*
*addition to the new materials designed in this work, the novelty of this work lies in the way*
*that the membrane filtration was driven by down-stream vacuum rather than a positive*
*pressure on the upstream side of the membrane. The manuscript is well-written and I read*
*the manuscript with interest as it offers a new energy-efficient and sustainable water*
*treatment process and may contribute to the broader scientific community. Hence, the*
*manuscript can be considered in Nature Communications after a revision, as some issues*
*require further clarifications. The detailed comments are listed below:*

1. *The method section described the fabrication methods: "To obtain the degummed silk*
*fibers, the raw silkworm silk was boiled in a 0.5 wt% Na₂CO₃ solution to remove sericin*
*proteins completely..." What is the composition of raw silk? I suggest that the authors*
*should provide a clear description of the structure of silk so that readers can*
*understanding more Information of silk nanofibers.*

**OUR RESPONSE:**

We appreciate the reviewer's comment. We have included the additional description in
the revised Supporting Information:

(Supporting Information: Page 3, Lines 122-128)

"Supplementary Fig. 2 briefly shows the fabrication process of SNF from the silkworm
fibers to the SNF suspension. The raw silk contains surface sericin covering the core
fibroin. When the surface sericin protein was removed, the fibroin can be further
processed into nano-size silk nanofibers (SNF)."

2. *In the manuscript, it is mentioned that "the SNF coating mass can alter the surface*
*morphology of the substrate". What were the morphology changes of SNF-NF*
*membranes with the additional SNF? More detailed characterization results should be*
*provided.*

**OUR RESPONSE:**

We appreciate the reviewer's comment. We have included the additional characterization
and relevant discussion in the revised Supporting Information:

(Supporting Information: Page 15, Lines 237-243)

"As shown in Supplementary Fig. 15, the surface area ratio and roughness (*R_a*) of SNF-
NF decreased with the additional SNF, which exhibited a similar tendency in
corresponding SNF-coated substrates."

Supplementary Fig. 15 | Morphology changes of the substrate-templated NF membranes with different SNF coating mass formed by IP reaction of 0.5 wt% PIP with 0.1 wt% TMC.

3. *Figure 2F shows a greater SNF-PIP intermolecular binding strength than PVDF-PIP. I suggest showing the relationship between interaction energy and binding strength more clearly and directly in Figure 2F.*

OUR RESPONSE:

We appreciate the reviewer's comment and have depicted the relationship between interaction energy and binding strength more clearly in revised **Figure 2F**.

Figure 2F | The total interaction energy of SNF-PIP and PVDF-PIP based on MD simulation results at 50 ns.

4. *In Fig. 1D, the later membrane demonstrated a higher surface area ratio. However, the control SNF0-NF0.5 exhibited higher water permeance than the SNF20-NF0.5 membranes. Please provide more convincing evidence and explanations for this experimental result.*

OUR RESPONSE:

We appreciated the reviewer's suggestions. The higher water permeance of SNF0-NF0.5 could be ascribed to the defects in their PA layers (**Supplementary Fig. 7**), allowing water molecules through the membranes rapidly through the defects. In addition, the low rejection of SNF0-NF0.5 may decrease the osmotic pressure difference between feed

and permeate, resulting in a highly effective driving pressure compared to that of SNF20-NF0.5.

Supplementary Fig. 7 | SEM images for the surface of SNF0-NF0.5 membranes with different resolution. Defects are marked by circles in these micrographs.

5. *Figure 4A illustrates the submerged vacuum-driven nanofiltration process. However, the construction of the filtration equipment is not entirely clear. I suggest the authors provide a digital photograph of the vacuum filtration apparatus to enhance understanding. Additionally, please clarify the installation of the NF membranes and methods for controlling vacuum pressure.*

OUR RESPONSE:

We appreciate the reviewer's comments and have included the information in the revised Supplementary Information.

(Supporting Information: Page 32, Lines 396-408)

"The digital photo shown in **Supplementary Fig. 32** demonstrates a vacuum filtration apparatus comprising a membrane cell immersed in a feed solution, a peristaltic pump providing suction to enable the vacuum-driven nanofiltration process, and an air valve connected to the vacuum system to regulate vacuum pressure. **Supplementary Fig. 33** shows the details of the membrane cell, containing a polyacrylic frame (connected to the suction peristaltic pump) and two stainless-steel splints. The NF membranes are sealed between the polyacrylic frame and stainless-steel splints using silicone rubber rings."

Supplementary Fig. 32 | Digital photo of the vacuum filtration apparatus.

Supplementary Fig. 33 | Digital photo of the membrane cell.

- 6. The authors changed the Na_2SO_4 concentration to 500 ppm in the vacuum-driven NF
 filtration test. Please explain why choosing such specific concentration which differs from
 many other studies using 1000 ppm concentration for their testing.

OUR RESPONSE:

We appreciate the reviewer's comments. For vacuum-driven nanofiltration, its maximum driving force is limited to 1 bar. Therefore, this process is more suitable to feed solutions

with relatively lower osmotic pressure. We chose 500 ppm in the vacuum-driven process, as the osmotic pressure is representative of that for municipal wastewater treatment (DOI: 10.1016/B978-0-08-096682-3.10002-2).

7. Figure 4C presents the separation performance of SNF20-NF0.1* and NF270 for five salts using a submerged vacuum-driven filtration process. What's the separation performance of these membranes in conventional cross-flow filtration mode.

OUR RESPONSE:

We appreciate the reviewer's comments. We have included the information in the revised Supplementary Information.

(Supplementary Information: Page 33, Lines 409-424)

"The separation performance of SNF20-NF0.1* and NF270 for five different salts have been tested in conventional cross-flow filtration mode (Supplementary Fig. 34 and 35). The SNF20-NF0.1* still maintained a high Na₂SO₄ rejection of > 96% and a high passage of essential minerals Ca²⁺ and Mg²⁺, resulting in improved minerals-sulfate selectivity compared to NF270."

Supplementary Fig. 34 | Separation performance of NF270 membranes for five different salts under the cross-flow filtration mode. The error bars represent the standard deviation of the salt rejection rate from three distinct samples ($n = 3$, testing condition: each test uses a single salt with a concentration of 1000 ppm at a hydraulic pressure of 3 bar).

Supplementary Fig. 35 | Separation performance of SNF20-NF0.1* membranes for

**five different salts under the cross-flow filtration mode.** The error bars represent the
standard deviation of the salt rejection rate from three distinct samples ($n = 3$, testing
condition: each test uses a single salt with a of 1000 ppm at a hydraulic pressure of 3 bar).

- 8. *The authors have calculated the specific energy consumption (SEC) based on previous*
*studies; however, some assumptions have changed. To ensure clarity and enable readers*
*to better understand the calculation process, the authors should describe the method*
*used for calculating SEC in this study.*

**OUR RESPONSE:**

We appreciate the reviewer's comments and have included the following information in
the revision:

(Manuscript: Page 24, Lines 531-535)

"Calculation of **specific energy consumption** and greenhouse gas emissions
**The specific energy consumption (SEC) for water purification is defined as the amount**
**of energy required to produce one cubic meter of purified water. In this study, we obtained**
**the SEC values using the method reported by the reference (DOI: 10.1021/acs.est.**
**0c05377), using the input parameters listed in **Supplementary Table 4.** In addition,** the
life cycle impact assessment of..."

**Supplementary Table 4. Operating parameters assumed in the calculation of SEC.**

Parameters	Value	
Mass transfer coefficient	k	100 L m ⁻² h ⁻¹
Transmembrane osmotic pressure difference	π_f	0.1 bar
Water recovery	Y	0.6
Average water flux	J _{av}	25 L m ⁻² h ⁻¹

9. *In your results, the SNF20-NF0.1* membranes achieved a PFOS removal of 99.6%, which*
*was considerably higher than the removal rates for other PFASs. Please explain this*
*significant difference.*

**OUR RESPONSE:**

We appreciated the reviewer's comments. PFOS has the highest molecular weight (M.W.
499 Da for PFOS ions) among these compounds. In addition, it has a negative charge
under the testing pH of 6.5. Due to electrostatic repulsion and steric hindrance, the
negatively charged SNF20-NF0.1* can achieve a higher removal rate for PFOS
compared to other PFASs used in this study.

- 10. *In Supplementary Figure 11, the water permeance is observed to decrease rapidly from*
*SNF20-PVDF to SNF50-PVDF. Please clarify the factors contributing to the decline in*

*water permeance.*

**OUR RESPONSE:**

We appreciated the reviewer's comments. **Supplementary Fig. 6** presents the influence
of varying SNF loading mass on the substrate morphology. With the increase of SNF
loading mass, the surface of PVDF was gradually coated from partial (SNF20-PVDF) to
fully covering (SNF50-PVDF) by the SNF. Compared with the SNF20-PVDF, the pores of
the pristine PVDF have diminished on the surface of SNF50-PVDF. In addition, we have
measured the thickness of SNF layer in the newly included **Supplementary Fig. 9**. The
thicker SNF layer could greatly increase the hydraulic resistance in the vertical (normal
to membrane) direction. Therefore, the combined effect of diminished surface pore and
thicker SNF layer leads to significantly increased hydraulic resistance and a rapid
decrease in water flux.

Supplementary Fig. 6 | Morphologies of the pristine and the SNF-coated PVDF substrates with different mass loading.

Supplementary Fig. 9 | Cross-sectional SEM micrographs of the control PVDF, SNF20-PVDF, SNF30-PVDF, and SNF50-PVDF substrates.

**Reviewer #2 (Remarks to the Author):**

*This paper presents nanofiltration membranes with ultrafast water permeance by using a SNF*
*gutter layer. The SNF layer preserves the rough topology of microfiltration supports while*
*absorbs enough amount of amine molecules at low concentrations, thereby facilitating*
*interfacial reaction for the creation of ultrathin nanofilms. The approach is novel, and the*
*performance is outstanding among literature reported membranes. Therefore, I would*
*suggest this paper to be considered for publication in Nature Communications, if the*
*comments below can be addressed.*

- 1. *The abstract should be restructured. The key advance in this paper is not the vacuum-*
*driven nanofiltration since it has not been proved as a practical alternative to the pressure-*
*driven nanofiltration in industry. Rather, the authors should shift the focus to the use of*
*SNF layer for enhancing the water permeance in nanofiltration, which is indeed the focus*
*throughout the main manuscript.*

**OUR RESPONSE:**

We agree with the reviewer that vacuum-driven NF has yet to be validated as a practical
industrial process. Nevertheless, this process has great potential for achieving great
savings in energy and cost. Our work could pave the way for the future practical
application of vacuum-driven NF.

In the main manuscript, we intended to focus on developing ultra-permeable NF
membranes that could potentially enable new process opportunities (vacuum-driven
submerged NF) in water treatment. To realize this, we then fabricated the SNF-
incorporated ultra-permeable NF membrane using a substrate-templated approach.
Likewise, we structured our introduction based on this rationale.

To further address the reviewer's comment, we have endeavored to accommodate the
reviewer's suggestions and make changes in the revised abstract.

(Manuscript: Page 2, Lines 25-33)

"... in such a system. Herein, we fabricated a silk-based membrane with a crumpled and
defect-free rejection layer, showing unprecedented water permeance of $96.2 \pm 10 \text{ L m}^{-2}$
$\text{h}^{-1} \text{ bar}^{-1}$ and a Na_2SO_4 rejection of $96.0 \pm 0.6\%$ under cross-flow filtration mode. In a
vacuum-driven system, ..."

"... greener water treatment process and paving the avenue for practical applications in
real industrial settings."

- 2. *The SNF layer should contribute to the membrane formation mechanism in three aspects:*
*i) preservation of the rough topology of MF supports; ii) enhanced absorption of PIP*
*solutions; and iii) interconnected porous structure with reduced tortuosity. However, the*
*authors only attributed the performance difference to the surface roughness, which could*
*be misleading. For example, SNF30-PVDF supports have similar surface roughness to*
*SNF20-PVDF supports (Supplementary Fig. 4), but the permeance of coated membranes*

*SNF30-NF0.5 is significantly lower than that of SNF20-NF0.5 (Fig. 3 and Supplementary*
*Fig. 10). This could be attributed to an enhanced tortuosity in thicker SNF layer (please*
*provide the thickness of SNF layer with various loading mass), rather than their difference*
*in surface roughness.*

**OUR RESPONSE:**

We agree with the reviewer's comment and have included these discussions the reviewer
mentioned in the revised manuscript and new **Supplementary Fig. 9** in the revised
Supplementary Information.

(Manuscript: Page 11-12, Lines 236-241)

"In addition to enhancing PIP loading, the SNF layer with an optimized thickness could
preserve the rough topology of MF supports to form a crumpled PA layer
(**Supplementary Fig. 6 and Supplementary Fig. 9**). Simultaneously, the interconnected
porous structure of the SNF layer could further reduce tortuosity on water transport,
resulting in significantly reduced hydraulic resistance. These combined effects
contributed to the enhanced separation performance of SNF-NF membranes³⁷."

(Manuscript: Page 13, Lines 268-271)

"Nevertheless, a further increase in SNF loading reduced the water permeance in SNF-
NF membranes (**Supplementary Fig. 21A**). This could be attributed to an enhanced
tortuosity in thicker SNF layer (**Supplementary Fig. 6 and Supplementary Fig. 9**) with
increased hydraulic resistance (**Supplementary Fig. 22**), highlighting the importance of
the optimized SNF loading mass."

Supplementary Fig. 6 | Morphologies of the pristine and the SNF-coated PVDF substrates with different mass loading.

Supplementary Fig. 9 | Cross-sectional SEM micrographs of the control PVDF,

**SNF20-PVDF, SNF30-PVDF, and SNF50-PVDF substrates.**

3. The enhanced absorption of PIP solutions enabled the fabrication of ultrathin nanofilms at
very low monomer concentrations (i.e., SNF20-NF0.2* and SNF20-NF0.1*), leading to
ultrahigh water permeance (Fig. 3). The active thickness of polyamide layer in these
membranes (i.e., SNF20-NF0.2* and SNF20-NF0.1*) should be measured to support this
phenomenal increase in their permeances.

**OUR RESPONSE:**

We measured the PA thickness in SNF20-NF0.2* and SNF20-NF0.1* by cross-section
TEM characterization. As shown in **Supplementary Fig. 23**, after reducing the IP
reaction and PIP concentration, the PA thickness of SNF20-NF0.1* was reduced to ~ 14
284 nm.

We have included the following sentences in the revised manuscript and new
**Supplementary Fig. 23** in the revised Supporting Information:

(Supporting Information, Page 23, Lines 319-325)

"The PA thickness in SNF20-NF0.2* and SNF20-NF0.1* have been measured based on
TEM cross-section images. As shown in **Supplementary Fig. 23**, after reducing the IP
reaction and PIP concentration, the PA thickness of SNF20-NF0.1* was reduced to
approximately 14 nm."

We have included the relevant discussion in the revised manuscript:

(Manuscript, Page 13, Lines 275-276)

"Notably, the SNF20-NF0.1* membrane with a thinner PA of approximately 14 nm
(**Supplementary Fig. 23**) demonstrates excellent water permeance of $96.2 \pm 10 \text{ L m}^{-2} \text{ h}^{-1}$
300 bar^{-1} with a Na_2SO_4 ."

**Supplementary Fig. 23** | Bright-field TEM images of the cross-section of SNF20-
NF0.5, SNF20-NF0.2* and SNF20-NF0.1*.

- 4. *Following the question above, thicker but defect-free polyamide layer should be fabricated*
*on the pristine PVDF (SNF0) support at higher PIP concentration. With the measured*
*thickness of active layer, it can further prove the advance of SNF layer for the creation of*
*ultrathin polyamide films.*

**OUR RESPONSE:**

We have included the following sentences and new **Supplementary Fig. 19** in the
revised Supporting Information:

(Supporting Information, Page 19, Lines 270-279)

“We have prepared the NF membranes with a higher PIP concentration of 1 wt% and 0.1
317 wt% TMC on the pristine PVDF support (SNF0-NF1). The PA layer was approximately
318 75 nm in thickness (**Supplementary Fig. 19**), which was much greater compared to
319 those for SNF0-NF0.5 (53 nm) and SNF20-NF0.5 (21 nm). For the membrane SNF20-
320 NF0.5, even though the SNF layer greatly increased the absorption of PIP, its PA
thickness remained relatively thin. This observation confirms the advantages of the SNF
layer for creating ultrathin PA films.”

**Supplementary Fig. 19 | Bright-field TEM images of the cross-section of SNF0-NF1.**
(IP condition: 0.1 wt% of TMC and 1 wt% of PIP; reaction time of the 60 s.)

5. *Line 273-276, the comparison and the conclusion is not convincing. Apart from the*
*difference in surface roughness, PES UF substrates should have a lower water*
*permeance (please provide this data) than PVDF MF substrates, which could significantly*
*contribute to the reduction of water permeance in the composite membranes.*

**OUR RESPONSE:**

We appreciate the reviewer’s comment. In this experiment, a smoother commercial **PES**
**MF substrate** was adopted instead of a UF substrate. The water permeance of MF
substrates has been measured. **Supplementary Fig. 24** demonstrated water permeance
of PES MF ($19360 \pm 1550 \text{ L m}^{-2} \text{ h}^{-1} \text{ bar}^{-1}$) is much higher than that of PVDF MF substrates
($7680 \pm 271 \text{ L m}^{-2} \text{ h}^{-1} \text{ bar}^{-1}$). Therefore, the reduction of water permeance in the composite

membranes may not be ascribed to the more permeable PES MF substrates, which
should be attributed to the much smoother polyamide layer.

We have included the following sentences and new **Supplementary Fig. 24** in the
revised Supporting Information:

(Supporting Information, Page 27, Lines 348-354)

**“Supplementary Fig. 27** demonstrated water permeance of PES MF ($19360 \pm 1550 \text{ L}$
$\text{m}^{-2} \text{h}^{-1} \text{bar}^{-1}$) is much higher than that of PVDF MF substrates ($7680 \pm 271 \text{ L m}^{-2} \text{h}^{-1} \text{bar}$
h^{-1}).”

We have revised the discussion in the manuscript:

(Manuscript, Page 13, Lines 283-285)

“To further prove the effectiveness of the crumpled structure, we coated the SNF layer
on a smoother MF substrate ($\sim Ra$ of 90 nm) with higher water permeance of $19360 \pm$
$1550 \text{ L m}^{-2} \text{h}^{-1} \text{bar}^{-1}$ (**Supplementary Fig. 27**). The resulting NF membrane had a much
lower water permeance ...”

**Supplementary Fig. 27** | The water permeance of PVDF MF and PES MF substrates.
The filtration test adopted a dead-end mode with an applied hydraulic pressure of 0.5 bar.
The error bars of water permeance represent the standard deviation of the test data from
three distinct samples ($n = 3$).

6. The upper bound in Fig. 3 needs to be updated. For example, the results from Ref. 20
should be added to the plot.

OUR RESPONSE:

We appreciate the reviewer's comment. **Figure 3B and 3C** have been revised to
include data from Reference 20.

Figure 3B | The trade-off relationship between the water permeance and water-salt ($A/B_{\text{Na}_2\text{SO}_4}$) selectivity of SNF20-NF0.1* and SNF20-NF0.2* membranes compared to other literature data.

Figure 3C | The trade-off relationship between the water permeance and NaCl/Na₂SO₄ selectivity of SNF20-NF0.1* and SNF20-NF0.2* membranes compared to literature data.

**Reviewer #3 (Remarks to the Author):**

*The authors tried to develop highly permeable nanofiltration (NF) membranes, which are*
*available for vacuum-driven NF, by spray-coating silk nanofiber (SNF) on a PVDF MF*
*membrane. However, the manuscript has many problems, as follows:*

*a) some interpretations are contradictory*

*b) some are misled into wrong interpretations*

*c) some suffer from the lack of empirical evidence or appropriate/fair comparisons*

*Therefore, the reviewer cannot agree with the publication of this manuscript in Nature*
*Communications. The details on the above comments are as follows.*

**Major comments**

**(1) Page 5, Lines 89-95**

*The SNF-incorporated NF membrane exhibited 96.2 LMH/bar while maintaining Na₂SO₄*
*rejection of 96% under cross-flow filtration, while it achieved water flux of 56.8 LMH and*
*Na₂SO₄ rejection of 96.3% at a partial vacuum of 0.9 bar. The reviewer raises an issue with*
*the similar Na₂SO₄ rejection despite the significant difference in the water flux (i.e., 96.2*
*LMH/bar vs. 56.8 LMH at a partial vacuum of 0.9 bar). In the two processes, the transport of*
*water molecules would be governed by hydraulic pressure, whereas ion transport should be*
*governed by ion concentration polarization. As such, since water molecules and ions move*
*by different driving forces, the significant difference in the water flux was likely to reduce*
*apparent Na₂SO₄ rejection on the condition that the feed solutions' concentrations were the*
*same. The authors need to think over this issue.*

**OUR RESPONSE:**

The reviewer has a good point. We address the reviewer's concern through the following
aspects:

**A. Feed concentrations used for cross-flow filtration and vacuum filtration**

We would like to clarify that different Na₂SO₄ concentrations were adopted in the two
filtration modes: 1000 ppm for cross-flow filtration and 500 ppm for vacuum filtration.
These concentrations have been included in the relevant figure captions. To avoid
potential misunderstanding, we have further revised **Fig. 4C** by clearly showing the
Na₂SO₄ concentration of 500 ppm in the figure.

In addition, we have included the following sentences to provide the rationale of using a
lower feed concentration for the vacuum-driven nanofiltration:

(Manuscript, Page 15, Lines 322-326)

"It is worthwhile to note, for vacuum-driven nanofiltration, its maximum driving force is
limited (< 1 bar). Therefore, this process is more suitable to feed solutions with relatively
low osmotic pressure. We chose a Na₂SO₄ concentration of 500 ppm in the vacuum-
driven process, as the osmotic pressure is representative of that for municipal
wastewater treatment (DOI: 10.1016/B978-0-08-096682-3.10002-2)."

B. Effect of operating conditions on rejection

We agree with the reviewer that the membrane rejection can be affected by operating conditions. Compared to the cross-flow filtration, the vacuum filtration in our study had a lower transmembrane pressure difference. Its lower water flux tends to reduce rejection due to reduced dilution effect (similar salt flux J_s but reduced water flux J_w , causing a higher permeate concentration [= J_s / J_w]). At the same time, lower water flux could also reduce the concentration polarization effect, which tends to improve rejection. Furthermore, the lower feed Na_2SO_4 concentration of 500 ppm adopted in the vacuum filtration tends to enhance rejection due to weakened charge screening effect. The similar Na_2SO_4 rejection values in the two testing modes are the compounded result of these competing effects.

To address the reviewer's concern and to allow more direct comparison of the two processes, we have further tested the separation performance of SNF20-NF0.1* membrane using a 1000 ppm Na_2SO_4 solution in vacuum filtration in the revised **Supplementary Fig. 36** and included the following sentences in the revised Supporting Information:

(Supporting Information, Page 34, Lines 426-439)

"We investigated the vacuum filtration performance of SNF20-NF0.1* membrane for a feed solution containing 1000 ppm Na_2SO_4 (**Supplementary Fig. 36**). The resulting Na_2SO_4 at 0.9 bar of $93.0 \pm 0.3\%$ was much lower compared to that of $96.0 \pm 0.6\%$ in the cross-flow filtration at 3 bar for the same feed solution. This difference could be attributed to the dilution effect: with a similar solute flux, a lower water flux results in greater permeate concentration and thus reduced salt rejection (DOI: 10.1016/j.memsci.2019.117297). In addition, the salt rejection for vacuum filtration of 1000 Na_2SO_4 was also lower than that of 500 Na_2SO_4 ($96.3 \pm 0.3\%$)."

Supplementary Fig. 36 | The separation performance of SNF20-NF0.1* under different operating conditions: (a) vacuum filtration at 0.9 bar for 500 ppm Na_2SO_4 ; (b) vacuum filtration at 0.9 bar for 1000 ppm Na_2SO_4 ; and (c) cross-flow filtration at 3 bar for 1000 ppm Na_2SO_4 . The error bars of separation performance represent the standard deviation of the test date from three distinct samples ($n = 3$).

To further resolve the different influencing factors, we have comprehensively investigated the influence of operational pressure and that of Na₂SO₄ concentration separately on the performance in two filtration modes.

B-1. Effect of applied pressure on rejection

We have included the following sentences and new **Supplementary Fig. 37** in the revised Supporting Information:

(Supplementary Information, Page 35, Lines 441-456)

"We investigated the influence of operational pressure on the separation performance in two filtration modes. **Supplementary Figure 37A** demonstrated that increasing the applied vacuum pressure can enhance the apparent Na₂SO₄ rejection in the vacuum-driven, which can be attributed to the dilution effect (DOI:10.1016/j.memsci.2019.117297). However, in cross-flow mode, the overly high operating pressure (such as 5 bar) may lead to severe concentration polarization that could jeopardize Na₂SO₄ rejection (**Supplementary Figure 37B**). The SNF20-NF0.1* demonstrated the best Na₂SO₄ rejection at the hydraulic pressure of 3 bar."

Supplementary Fig. 37 | The influence of operational pressure on the membrane separation performance of SNF20-NF0.1* in two processes (A) Separation performance of SNF20-NF0.1* with different vacuum pressures under the vacuum-driven mode (feed solution of 500 ppm Na₂SO₄). (B) Separation performance of SNF20-NF0.1* with different hydraulic pressure under the cross-flow mode (feed solution of 1000 ppm Na₂SO₄). The error bars represent the standard deviation of the salt rejection rate from the measurement data of three distinct samples.

B-2. Effect of feed concentration on rejection

We have included the following sentences and new **Supplementary Fig. 38** in the revised Supporting Information:

(Supplementary Information, Page 36, Lines 457-469)

"We investigated the influence of Na₂SO₄ concentrations (500 ppm vs. 1000 ppm) in the

498 two filtration modes. **Supplementary Fig. 38** demonstrated that increasing feed
 concentrations decreased the Na_2SO_4 rejections in both cross-flow and vacuum-driven
 filtration modes.”

**Supplementary Fig. 38** | The influence of the feed concentration of Na_2SO_4 on the
 membrane separation performance of SNF20-NF0.1*. (A) Separation performance of
 SNF20-NF0.1* with different Na_2SO_4 concentrations under the vacuum-driven mode
 (hydraulic pressure of -0.9 bar). (B) Separation performance of SNF20-NF0.1* with
 different Na_2SO_4 concentrations in the cross-flow mode (hydraulic pressure of 3 bar). The
 error bars represent the standard deviation of the salt rejection rate from the
 measurement data of three distinct samples.

(2) Fig. 1(c) and 1(d)

The authors need to provide the roughness data of the control PVDF MF membrane's surface.

**OUR RESPONSE:**

We have included the R_a value of PVDF MF in the revised manuscript for a clearer
 understanding.

(Manuscript: Page 7, Lines 140-141)

“At the same time, their roughness values are comparable to that of the substrate ($R_a =$
 388 nm, **Supplementary Fig. 6A**).”

**Supplementary Fig. 6A** | Morphologies of the pristine PVDF substrates.

(3) Page 7, Lines 119-123

According to conventional wisdom, organic materials smaller and larger than 0.45 μm are
considered dissolved organic matter (DOM) and particulate matter, respectively. In this regard,
SNF, which was 380 nm in length, corresponds to DOM, indicating that the authors spray-
coated DOM on the MF membrane's surface without chemical crosslinking. Can the authors
guarantee SNF is not leaked to a permeate solution during filtration?

OUR RESPONSE:

We appreciate the reviewer for raising this critical point regarding SNF leakage during
filtration experiments. To address the reviewer's concern, we have included following
discussion in revised Supplementary information:

(Supporting Information: Page 5-6, Lines 143 -167)

"We have designed experiments to examine the stability of the SNF coating layer.
Specifically, an SNF-coated PVDF substrate was immersed in TMC-hexane solution for
1 minute and then vigorously stirred in DI water for 10 minutes. This immersion-stirring
process was repeated for three times. For comparison, another SNF-coated PVDF
substrate was directly stirred in DI water without treatment with a TMC-hexane solution.
We further compared the surface morphology change of these substrates by SEM
characterization (**Supplementary Fig. 4**). The SEM characterization revealed that the
SNF coating treated with TMC solution remained complete on the substrate after washing,
whereas the control SNF coating was washed away. FTIR results (**Supplementary Fig.**
**5**) also validated that the characteristic SNF peak persisted on TMC-treated coating
layers following water washing, demonstrating the excellent stability of SNF coating
layers. This improved stability may be caused by the crosslinking of SNF by TMC through
the reaction of the abundant amine groups of SNF with the acyl chloride groups of TMC."

**Supplementary Fig. 4 | SEM images of the surface morphology of substrates with**
**different treatment. (A) Pristine PVDF substrate (B) SNF-coated substrate (C) SNF-**

coated substrate washed by DI water (D) SNF-coated substrate treated with TMC solution, then washed by DI water.

Supplementary Fig. 5 | The FTIR results of substrates with different treatments.

(4) Page 8, Lines 145-149

The authors insisted that the hydrophilic nature of SNF may be beneficial for creating favorable membrane formation conditions, while the dynamic contact angle results demonstrated better water distribution ability of the SNF-coated substrate based on Supplementary Fig. 6. However, according to Supplementary Fig. 6, the contact angle values of the control PVDF and the SNF20-PVDF membrane were approaching 0. Accordingly, the contact angle data of the SNF20-PVDF membrane cannot be used to corroborate the hydrophilic nature of SNF. The two contact angle data were highly likely to be associated with highly wetting surfaces stemming from the highly porous surface of the two MF membranes rather than their hydrophilicity.

OUR RESPONSE:

The reviewer raised a good point that the contact angle results may not demonstrate a better water distribution ability of the SNF-coated substrate. To address the reviewer's concern, we decided to remove the relevant discussions (Page 8, Line 147-149 in the original manuscript: "~~Indeed, the dynamic contact angle results demonstrated better water distribution ability of the SNF-coated substrates (Supplementary Fig. 6), which further helped to uniformly distribute PIP solution during the IP reaction.~~"). This minor change does not affect the overall integrity of the manuscript.

In addition, we have conducted additional experiments to demonstrate the hydrophilic nature of SNF coatings and revised the relevant discussion in the manuscript based on

the reviewer's constructive comment.

(Supporting Information: Page 11-12, Lines 197-218)

**“Supplementary Fig. 12** shows the dynamic contact angle results of PVDF and SNF20-
PVDF. For both substrates, the contact angle approached zero at time longer than
approximately 10s, which is likely to due to the highly porous surface of the two substrates.
Nevertheless, the SNF20-PVDF had a faster decrease in the contact angle values. To
avoid interference from the highly porous surface of the substrates, we coated SNF on a
polysulfone (PSF) UF membranes with small surface pore size and a low surface porosity
**(Supplementary Fig. 10)** for water contact angle measurement (time = 10s). As shown
in **Supplementary Fig. 11**, compared with the pristine PSF, SNF-coated PSF substrates
exhibited a significantly improved surface hydrophilicity with a lower contact angle of
60.7°, compared to the pristine PSF substrate with a contact angle of 80.2°. This
observation suggested the SNF coating imparted increased hydrophilicity to the substrate
surface.”

(Manuscript: Page 8, Lines 150-153)

Revised discussion: **“Furthermore, the hydrophilic nature of SNF coating**
**(Supplementary Fig. 10 and 11)** and highly wetting surfaces of SNF-coated substrates
**(Supplementary Fig. 12)** may create favorable membrane formation conditions, leading
to enhanced quality of the generated PA rejection layer.”

Supplementary Fig. 10 | Surface SEM images of substrates.

(A) Pristine PSF: $80.2 \pm 1.4^\circ$

(B) SNF-coated PSF: $60.7 \pm 1.1^\circ$

Supplementary Fig. 11 | The contact angle results of pristine and SNF-coated PSF

**substates.**

**Supplementary Fig. 12 | The dynamic water contact angle of the pristine PVDF,**
**SNF20-PVDF substrates, and SNF20-NF0.5 membranes.**

**(5) Page 10, Lines 189-205**

*The authors presumed that the crosslinking degrees of the SNF-NF membranes with SNF*
*were higher than the SNF0-NF0.5 membrane based on their lower O/N ratio and lower*
*negative surface zeta potential. However, the O/N ratio and the surface zeta potential cannot*
*directly support the higher crosslinking of the SNF-NF membranes with SNF due to the*
*following reasons. First, SNF also has a lot of nitrogen in its repeating unit. Thus, the authors*
*cannot conclude that the SNF-NF membrane with SNF showed higher crosslinking degree*
*without consideration of the nitrogen content contribution of SNF. Second, SNF is known to*
*posses more OH groups than carboxyl groups, which is stark contrast to polyamide*
*containing much more carboxy groups than hydroxyl groups. Accordingly, SNF itself can*
*endow the SNF-NF membranes containing SNF with lower negative surface zeta potential*
*regardless of crosslinking degree of polyamide. Lastly, the authors attributed the high O/N*
*ratio of the SNF0-NF0.5 membrane to the extra oxygen content of the PVDF substrate.*
*However, considering the polyamide film's thickness (i.e., ca. 20 nm) and the detection depth*
*of XPS, the O/N ratio of the SNF0-NF0.5 is thought not to arise from the PVDF substrate's*
*oxygen content.*

**OUR RESPONSE:**

We appreciate the reviewer's comment. To address reviewer's concerns, we have
conducted additional experiments and revised the corresponding discussions in the
manuscript. To more systematically address the reviewer's concerns, we decide our
response into the following 3 parts:

**Reviewer Comment 5-1:** "First, SNF also has a lot of nitrogen in its repeating unit. Thus, the
authors cannot conclude that the SNF-NF membrane with SNF showed higher crosslinking
degree without consideration of the nitrogen content contribution of SNF."

**Response 5-1:** The reviewer raised a good point that SNF coatings may influence on
calculating the O/N ratios of PA layers and then interfere with the comparison of their

cross-linking degree. To address the reviewer's concern, we decided to remove the
relevant discussion on the O/N ratio. Instead, we have conducted additional experiments
to demonstrate the influence of SNF on the pore size of the PA and revised the relevant
discussion in manuscript:

Deleted discussion

(Page 10, Lines 196-198, in the original manuscript): "In addition, SNF-NF membranes
with the increased SNF loading exhibited a gradually decreased O/N ratio (from $1.13 \pm$
0.03 of SNF20-NF0.5 to 1.03 ± 0.02 of SNF50-NF0.5), corresponding to a more cross-
linked PA nanofilm."

Added discussion

(Page 10, Lines 200-206 in revised manuscript): To further clarify the influence of SNF
on the properties of the PA, we employed Doppler Broadening Energy Spectroscopy
(DBES) to compare free volume (or the size of sub-nanometer pores) in the PA layers of
SNF0-NF0.5 (without SNF) and SNF20-NF0.5 (with SNF) membranes. Typically, a
smaller S parameter of DBES indicates a lower free volume. As shown in **Supplementary**
**Fig. 16**, SNF20-NF0.5 possessed a smaller S value than SNF0-NF0.5, revealing the
formation of a denser PA layer in the presence of the SNF coating."

**Supplementary Fig. 16** | The Doppler broadening energy spectroscopy results of
SNF0-NF0.5 and SNF20-NF0.5.

**Reviewer Comment 5-2:** "Second, SNF is known to possess more OH groups than carboxyl
groups, which is stark contrast to polyamide containing much more carboxy groups than
hydroxyl groups. Accordingly, SNF itself can endow the SNF-NF membranes containing SNF
with lower negative surface zeta potential regardless of crosslinking degree of polyamide."

**Response:** We appreciate the reviewer's insightful comment. In the revision, we have
deleted the related discussion about the possible connection between zeta potential
results and the XPS results. This minor change does not affect the overall discussion in
the manuscript.

(Lines 198 – 205, Page 10, in the original manuscript: “Meanwhile, the zeta potential
results show that SNF20-NF0.5 membrane with SNF coating had a less negative surface
charge than the control NF membrane (Fig. 2C). This observation may also corroborate
a more cross-linked PA formed in the SNF20-NF0.5 membrane (i.e., less hydrolysis of
acryl chloride groups in TMC during IP reaction). Indeed, the improved compatibility and
hydrophilicity of the SNF coating layer can favor the formation of the PA rejection layer,
which is further elaborated in the following section: “Role of SNF in the IP Process.”).

In addition, we revised the discussion about the zeta potential of SNF0-NF0.5 and
SNF20-NF0.5 in the revised manuscript:

(Manuscript: Page 10, Lines 206 - 209)

“In addition, we investigated the surface zeta potential of SNF0-NF0.5 and SNF20-NF0.5
membranes (Fig. 2C). The results demonstrated that both membranes have a negatively
charged membrane surface over a wide pH range, which can contribute to the rejection
of charged solutes based on the Donnan effect.”

**Reviewer Comment 5-3:** Lastly, the authors attributed the high O/N ratio of the SNF0-NF0.5
membrane to the extra oxygen content of the PVDF substrate. However, considering the
polyamide film's thickness (i.e., ca. 20 nm) and the detection depth of XPS, the O/N ratio of
the SNF0-NF0.5 is thought not to arise from the PVDF substrate's oxygen content.

**Response:** The reviewer raised an excellent point. We agree that XPS had limited
sample penetration depth (i.e., ca. 20 nm). However, the SNF0-NF0.5 membrane
contains numerous defects (Supplementary Fig. 7), which allows the partial exposure
of substrate material to XPS detection. This point has been reflected in the original
manuscript:

(Manuscript: Page 10, Lines 198-200)

“The unusual O/N ratio of SFN0-NF0.5 could result from the extra oxygen contents from
substrates (Supplementary Table 1), which can be detected by XPS through the defect
regions of PA.”

Supplementary Fig. 7 | SEM images for the surface of SNF0-NF0.5 membranes with different resolution. Defects are marked by circles in these micrographs.

**(6) Fig. 3(a)**

*Based on Fig. 3(a), the reviewer thinks that the effects of reducing interfacial polymerization*
 *(IP) time and PIP content were much more dominant than that of SNF coating. The authors*
 *should provide data regarding the water permeance and Na₂SO₄ rejection of the SNF0-*
 *NF0.2* and SNF0-NF0.1* to clarify the dominant factor influencing the filtration performance.*
 .

OUR RESPONSE:

As suggested, we have tested the separation performance of the SNF0-NF0.2* and SNF0-NF0.1* NF membranes to clarify the dominant factor influencing the filtration performance. As shown in **Supplementary Fig. 24**, both SNF0-NF membranes without SNF coating exhibited much lower Na₂SO₄ rejection (< 40%) compared to the corresponding SNF-incorporated SNF-NF membranes (**Fig. 3A**), confirming that the SNF coatings played a dominant role in forming an intact membrane and maintaining good rejection. Meanwhile, in the presence of SNF coatings, reducing interfacial polymerization (IP) time and PIP concentration can further enhance the water permeance of NF membranes, without severely impacting Na₂SO₄ rejection (> 96% for all PIP concentrations of 0.1 ~ 0.5 wt%). As further discussed in our Response #7, the SNF coating with high PIP affinity provides a sufficient supply of PIP monomers, which helps to minimize defect formation. Therefore, the combination of the SNF layer and optimized IP conditions contributed to the excellent performance of SNF20-NF0.1*.

We have included the following sentences and new **Supplementary Fig. 24** in the

revised Supporting Information:

(Supporting Information: Page 24, Lines 326-334)

“As shown in **Supplementary Fig. 24**, both SNF0-NF membranes without SNF coating exhibited much lower Na_2SO_4 rejection (< 40%) compared to the corresponding SNF-incorporated SNF-NF membranes (**Fig. 3A**), confirming that the SNF coatings played a dominant role in improving rejection.”

Supplementary Fig. 24 | The separation performance of SNF0-NF0.1* and SNF0-NF0.2*. The filtration test adopted a conventional pressure-driven cross-flow mode with an applied hydraulic pressure of 3 bar. The rejection test was performed using a feed solution of 1000 ppm Na_2SO_4 and pure water permeance was determined using DI water.

(7) Page 11, Lines 215-234

The authors insisted that the SNF coating layer can enhance the loading of PIP monomers onto the template substrate, subsequently enhancing the IP reaction to form a better-quality PA nanofilm. However, Fig. 3(a) displayed that the lower content of PIP led to higher water permeance while maintaining comparable Na_2SO_4 rejection. The water permeance data shown in Fig. 3(a) looks contradictory to the above authors' claim, considering that the lower content of PIP should result in the lower loading of PIP onto the template substrate during IP. The authors should address this contradictory issue in their interpretation.

OUR RESPONSE:

We thank the reviewer for raising this critical point. To address the reviewer's concern, we have added the following to the manuscript

(Manuscript, Page 17-18, Lines 379-392):

“It is worthwhile to note that, for conventional interfacial polymerization, decreasing PIP concentration is often adopted in order to enhance water flux. Nevertheless, this primitive strategy greatly increases the risks of forming more defects in the PA rejection layer (1. DOI: 10.1016/j.memsci.2021.119450; 2. DOI: 10.1016/j.memsci.2017.09.046; 3. DOI: 10.1016/j.seppur.2019.01045). The tendency of defect formation is largely due to insufficient supply of PIP monomers (DOI: 10.1021/acs.iecr.3c02779). In contrast, the novel SNF layer offers major advantages over the conventional approach. SNF increases

the sorption of PIP, which allows it to act as a “PIP reservoir” to ensure sufficient supply
of amine monomer (**Fig. 2D**). At the same time, the SNF layer provides a more defined
reaction interface with slower release of the PIP monomer (**Supplementary Fig. 20**).
Due to these combined effects, the SNF achieves simultaneously moderate effective PIP
concentration at the IP reaction interface (which is favorable for forming a more
permeable NF membrane) and sufficient PIP storage (which minimizes defect formation;
DOI: 10.1021/acs.est.8b02425). In addition, the large PIP storage in SNF, together with
reduced surface pore size of SNF-coated substrates, allows the adoption of lower PIP
concentrations for IP reaction while still maintaining high rejection of Na₂SO₄ (**Fig. 3A**).

(Supporting Information: Page 20, Lines 280-290)

“We further performed QCM-D tests to investigate the influence of SNF on the desorption
behavior of PIP monomers. Specifically, a quartz crystal sensor with or without SNF
coating was mounted into the QCM-D chambers. In order to achieve an identical initial
loading of PIP (approximately 1500 ng cm⁻²), we used a 8 wt % PIP solution for the control
sensor without SNF and a 0.5 wt % PIP solution for the SNF-coated sensor. After
stabilization, DI water was introduced into the chambers to determine the kinetics of PIP
desorption. As shown in **Supplementary Fig. 20**, the SNF coating could significantly
reduce the released rate of PIP.”

**Supplementary Fig. 20** | Quartz crystal microbalance with dissipation (QCM-D)
release test of PIP monomer for a control sensor without SNF coating and a SNF-
coated sensor.

(8) Fig. 4(b) and 4(e)

The authors compared the water flux of the SNF20-NF0.1* only with the NF270, which should
be prepared with a different type of substrate (e.g., a dense ultrafiltration membrane). Hence,
the authors need to provide the water flux of the SNF0-NF0.1* for a fair comparison and
further verification of the SNF coating effect. It is also the case for the comparison of the GHG
emissions. In the comparison of the GHG emissions, the data regarding the SNF0-NF0.1*
should be provided in parallel with the SNF20-NF0.1*.

**OUR RESPONSE:**

Following the reviewer's comment, we have added SNF0-NF0.1* in Fig. 4B. Both the
flux and Na₂SO₄ rejection under vacuum filtration were reported in the revised Fig. 4B.
The following discussion is added to the revision:

(Manuscript Page 15, Lines 329-333)

"On the other hand, even though the SNF0-NF0.1* membrane without SNF had higher
flux compared to the SNF20-NF0.1* membrane, the former exhibited a low Na₂SO₄
rejection of merely 21%. Without the SNF coating, the high tendency of forming defects
in the PA layer (Supplementary Fig. 7) makes this membrane less suitable for NF
applications."

**Fig. 4 (B)** The comparison of separation performance among commercial NF270
membranes, SNF20-NF0.1* and SNF0-NF0.1* under vacuum filtration using a feed
solution of 500 ppm Na₂SO₄ (vacuum pressure of -0.9 bar).

In our study, the SNF0-NF0.1* membrane without SNF coating has low Na₂SO₄ rejection
and is not suitable for NF applications. Therefore, we decide not to include this membrane
for the GHG emission comparison in Fig. 4E.

**Minor comments**

1) Page 8, Line 142

It can be difficult for readers to understand the meaning of SNF20-NF0.5. Accordingly, it is
advisable to briefly explain the meaning of the sample names in the Methods.

**OUR RESPONSE:**

We appreciate the reviewer's comment. We have briefly explained the meaning of the

sample names in the revised Methods and updated the membrane fabrication conditions
in **Supplementary Table 4**.

(Manuscript: Page 20, Lines 427-430)

“SNF-NF membranes were stored in water at 4°C for further characterization. More
details regarding the IP conditions and the corresponding sample names can be found
in **Supplementary Table 4**. For example, SNF20-NF0.5 stands for the NF membrane
fabricated on a substrate (63.5 cm²) coated with a volume of 20 mL SNF (129 µg mL⁻¹)
using a 0.5 wt% PIP solution in IP reaction.”

**Supplementary Table 4**. The fabrication recipe for SNF-NF membranes with and without
SNF.

NF membranes	Substrates	PIP concentration (wt%)	TMC concentration (wt%)	IP time (s)
SNF0-NF0.5	PVDF	0.5	0.1	60
SNF20-NF0.5	SNF20-PVDF	0.5	0.1	60
SNF30-NF0.5	SNF30-PVDF	0.5	0.1	60
SNF50-NF0.5	SNF50-PVDF	0.5	0.1	60
SNF0-NF0.1*	PVDF	0.1	0.1	30
SNF0-NF0.2*	PVDF	0.2	0.1	30
SNF20-NF0.1*	SNF20-PVDF	0.1	0.1	30
SNF20-NF0.2*	SNF20-PVDF	0.2	0.1	30

*2) The authors need to provide the mean pore size of the PVDF MF membrane.*

**OUR RESPONSE:**

We have included the following information in the revised manuscript and new
**Supplementary Fig. 8** in the revised Supporting Information:

(Manuscript: Page 8, Lines 144-145)

“...addition, due to the large pore size of the MF substrate (mean pore size of 0.218 µm,
**Supplementary Fig. 8**), the formed thin PA rejection layer has to span over the large
pore region.

Supplementary Fig. 8 | The pores distribution of PVDF MF substrate. The pore size distribution was measured using a Pore Size Analyzer (BSD-660S, BSD Instrument Co., Ltd).

Reviewers' Comments:

Reviewer #1:

Remarks to the Author:

The authors properly addressed reviewer's concerns and the revised work is recommended for publication

Reviewer #2:

Remarks to the Author:

The authors have addressed all of my comments. I would suggest the manuscript for the publication in Nature Communications.

Reviewer #3:

Remarks to the Author:

Recommendation: Major revision

The authors tried to answer the reviewer's comments. However, the revised manuscript still has critical problems in the revised manuscript. Thus, the following issues should be addressed.

1. Supporting Information: Page 5-6, Line 143-167

The authors tested the as-prepared membranes' stability based on the SEM images and FTIR spectra obtained after immersing SNF-coated substrates in a TMC solution for 1 min, followed by immersing them in DI water for 10 min during stirring. However, SEM images and FTIR spectra provide only qualitative data obtained from a very small area. Accordingly, it is undesirable to discuss the SNF-coated NF membrane's stability based on the SEM image and FTIR spectra of even SNF-coated substrates. Furthermore, stirring for 10 min is not enough to evaluate the membrane stability. Lastly, please note that the reviewer had raised an issue with the leakage possibly occurring during filtration due to no chemical bonding or crosslinking between silk fibroins and substrates. Overall, the authors should provide the water flux and salt rejection data (i.e., quantitative data) obtained while performing the continuous long-term filtration test of the as-prepared NF membranes for at least two days to one week. The authors are able to demonstrate the SNF-coated membrane's stability by representing that the SNF-coated NF membrane maintains filtration performance in terms of water permeability and salt rejection.

2. Page 10, Line 198-200 & Supplementary Fig. 7

The author insisted that the unusual O/N ratio of SFN0-NF0.5 could result from the extra oxygen contents from substrates, which could be detected by XPS through the defect regions of PA. First of all, the substrate was a PVDF membrane, which seems to be far from the author's claim that the extra oxygen contents resulted from substrates. If the authors attributed the extra oxygen contents to the PVDF substrate because it was a hydrophilic PVDF membrane, the authors should have

provided the chemical composition of the PVDF substrate at least. Second, the reviewer doubts whether it is technically possible to aim to detect such a small area of the 100-nm defect by XPS. If the authors intended only the detection inclusive of defects, the authors need to note that the area proportion of defects is negligible compared to the entire observed area. It looks appropriate to find another way to prove that SNF-NF membranes had similar crosslinking degree or density to other TFC NF membranes reported in the literature.

3. Fig. 4(c)

The SNF-NF membrane showed much higher rejection for Na₂SO₄(96%) than MgSO₄(about 50%), which is far from the literature. In general, MgSO₄ rejection is higher than or similar to Na₂SO₄ rejection. For your reference, a couple of papers are provided below.

[1] N.P. i Hernando, Nanofiltration and hybrid sorption: ultrafiltration processes for improving water quality, in, Universitat Politècnica de Catalunya, 2017.

[2] A.S. Colburn, Synthesis, Functionalization, and Application of Nanofiltration and Composite Membranes for Selective Separations, (2019).

The authors should elaborate on why the SNF-NF membrane exhibited a significantly different tendency in the rejection rates of Na₂SO₄ and MgSO₄ from the literature.

Response to the comments:

Reviewer #1 (Remarks to the Author):

The authors properly addressed reviewer's concerns and the revised work is recommended for publication

OUR RESPONSE:

We thank you for your recommendation for publication.

Reviewer #2 (Remarks to the Author):

The authors have addressed all of my comments. I would suggest the manuscript for the publication in Nature Communications.

OUR RESPONSE:

We thank you for your recommendation for publication.

Reviewer #3 (Remarks to the Author):

Recommendation: Major revision.

The authors tried to answer the reviewer's comments. However, the revised manuscript still has critical problems in the revised manuscript. Thus, the following issues should be addressed.

OUR RESPONSE:

We appreciate the reviewer's comments and endeavor to address his/her concerns.

Comment #1. Supporting Information: Page 5-6, Line 143-167

The authors tested the as-prepared membranes' stability based on the SEM images and FTIR spectra obtained after immersing SNF-coated substrates in a TMC solution for 1 min, followed by immersing them in DI water for 10 min during stirring. However, SEM images and FTIR spectra provide only qualitative data obtained from a very small area. Accordingly, it is undesirable to discuss the SNF-coated NF membrane's stability based on the SEM image and FTIR spectra of even SNF-coated substrates. Furthermore, stirring for 10 min is not enough to evaluate the membrane stability. Lastly, please note that the reviewer had raised an issue with the leakage possibly occurring during filtration due to no chemical bonding or crosslinking between silk fibroins and substrates. Overall, the authors should provide the water flux and salt rejection data (i.e., quantitative data) obtained while performing the continuous long-term filtration test of the as-prepared NF membranes for at least two days to one week. The authors are able to demonstrate the SNF-coated membrane's stability by representing that the SNF-coated NF membrane maintains filtration performance in terms of water permeability and salt rejection.

OUR RESPONSE:

We appreciate the reviewer's comment. To address the reviewer's concern, we have conducted the long-term cross-flow filtration test of SNF20-NF0.1* and included the following sentences and new **Supplementary Fig. 39** in the revised Supporting Information:

(Supporting Information: Page 38, Lines 482-495)

To further confirm the stability of the SNF20-NF0.1*, we tested its separation performance under cross-flow condition over a period of 7 days using a feed solution of 1000 ppm Na₂SO₄ at a hydraulic pressure of 3 bar (**Supplementary Fig. 29**). The membrane maintained a stable Na₂SO₄ rejection of > 96%. The water flux was slightly reduced, which is likely due to membrane compaction^{1, 2}. The stable Na₂SO₄ rejection implies a good stability of the membrane.

Supplementary Fig. 39 | The long-term separation performance of SNF20-NF0.1*. The rejection test was performed using a feed solution of 1000 ppm Na₂SO₄. The filtration tests adopted a pressure-driven cross-flow mode with an applied hydraulic pressure of 3 bar. The error bars of separation performance represent the standard deviation of data from three distinct samples (n = 3).

Comment #2. Page 10, Line 198-200 & Supplementary Fig. 7

The author insisted that the unusual O/N ratio of SFN0-NF0.5 could result from the extra oxygen contents from substrates, which could be detected by XPS through the defect regions of PA. First of all, the substrate was a PVDF membrane, which seems to be far from the author's claim that the extra oxygen contents resulted from substrates. If the authors attributed the extra oxygen contents to the PVDF substrate because it was a hydrophilic PVDF membrane, the authors should have provided the chemical composition of the PVDF substrate at least. Second, the reviewer doubts whether it is technically possible to aim to detect such a small area of the 100-nm defect by XPS. If the authors intended only the detection inclusive of defects, the authors need to note that

the area proportion of defects is negligible compared to the entire observed area. It looks appropriate to find another way to prove that SNF-NF membranes had similar crosslinking degree or density to other TFC NF membranes reported in the literature.

OUR RESPONSE:

We appreciate the reviewer's insightful comments. To systematically address the reviewer's concerns, we divide our response into the following 3 parts:

Reviewer Comment 2-1: *“First of all, the substrate was a PVDF membrane, which seems to be far from the author's claim that the extra oxygen contents resulted from substrates. If the authors attributed the extra oxygen contents to the PVDF substrate because it was a hydrophilic PVDF membrane, the authors should have provided the chemical composition of the PVDF substrate at least.”*

Response 2-1: Oxygen-riched PVDF substrate and its chemical composition

As stated in the methodology (Supporting Information: Page 1, Lines 113-115), we adopted a commercially available hydrophilic PVDF substrate (GVWP09050, Merck Millipore).

To address the reviewer's comment, we performed XPS analysis of the bare PVDF substrate. The XPS results confirmed high oxygen content of the PVDF substrate (15.92%, revised **Supplementary Table 1**), which may be attributed to hydrophilic modification although the exact chemical composition is proprietary.

Supplementary Table 1. Elemental compositions of the top surface of PVDF, SNF-PVDF and SNF0-NF0.5 measured by XPS.

The XPS results	Atom percent (%)				
	O 1s	N 1s	C 1s	F 1s	O/N
Pristine PVDF	15.92	N.D. ^a	59.05	25.03	N.A. ^b
SNF20-PVDF	17.14	7.69	60.60	14.57	2.23
SNF30-PVDF	17.36	6.54	60.72	15.38	2.66
SNF50-PVDF	20.06	14.54	62.64	2.76	1.38
SNF0-NF0.5 (control)	11.27	4.61	63.38	20.74	2.44

Notes: a. Not detected. b. Not applicable.

Reviewer Comment 2-2: “Second, the reviewer doubts whether it is technically possible to aim to detect such a small area of the 100-nm defect by XPS. If the authors intended only the detection inclusive of defects, the authors need to note that the area proportion of defects is negligible compared to the entire observed area.”

Response 2-2: Unusual O/N ratio of SFN0-NF0.5

A. Further explain the unusual O/N could be attributed to the partial exposure of oxygen-riched PVDF substrate via defects in the PA

- The reviewer has raised a good point about the small proportion of defect areas in the SEM micrographs. We would like to highlight that SEM sample preparation involves sputter coating by Au and Pt of approximately 5 nm³, which may seal some small defective regions. Furthermore, the limited resolution of SEM means small defects of a few nanometers are difficult to be detected. Therefore, the actual proportion of defective region in the PA could be larger than that of observed from the SEM images. These points have been added to the revised discussion for **Supplementary Fig. 7**:

(Supporting Information: Page 8, Lines 185-190)

Supplementary Fig. 7 showed that defects appeared on the surface of SNF0-NF0.5 membranes (control NF membranes) in different resolutions. It is worthwhile to note that SEM sample preparation involves sputter coating by Au and Pt of approximately 5 nm³, which may seal some small defective regions. Furthermore, the limited resolution of SEM means small defects of a few nanometers are difficult to be detected. Therefore, the actual proportion of defective regions in the PA could be larger than that of observed from the SEM images.

- The XPS result of SNF0-NF (**Supplementary Table 1**) showed a high F content of ~20.74%, reflecting the partial exposure of PVDF substrate in XPS detection.

B. PA hydrolysis

- An additional plausible cause for the very high O/N ratio might be the hydrolysis of PA. Indeed, O/N ratios of >2 have been reported in the membrane community. According to a recently published paper by Xu and coworkers (Nature Communications; DOI:10.1038/s41467-024-45918-4)⁴, an unfavorable IP reaction could result in an increased number of PA oligomers, resulting in O/N ratio between 2 – 5.
- In our work, the insufficient supply of PIP for the SFN0-NF0.5 membranes may promote the excessive hydrolysis of TMC, resulting in an increased number of PA oligomers. These could result in more

residue of carboxyl groups in the PA network, leading to an O/N ratio of >2.

- In contrast, the SNF layer provides a more defined interface/sufficient PIP storage, which results in optimized PA layer formation with less formation of PA oligomers.

To further address the reviewer's comment, we have included the following discussions in the revised manuscript:

(Manuscript: Page 10, Lines 198-203)

“The unusual O/N ratio of SFN0-NF0.5 could result from the extra oxygen contents. The additional oxygen contents could be possibly attributed to two sources. First, the hydrolysis of TMC in the IP reaction and the formation of PA oligomers could produce additional carboxyl groups in the PA network⁴. In addition, the oxygen-riched substrates (**Supplementary Table 1, XPS result**) could be detected by XPS through the defect regions of PA (**Supplementary Fig. 7**).”

Reviewer Comment 2-3: “It looks appropriate to find another way to prove that SNF-NF membranes had similar crosslinking degree or density to other TFC NF membranes reported in the literature.”

Response 2-3:

A. Doppler Broadening Energy Spectroscopy (DBES)

To further address the reviewer's comment, we have included a new **Supplementary Table 5** and the following discussions in the revised Supporting Information:

(Supporting Information: Page 41, Lines 509-519)

“Doppler Broadening Energy Spectroscopy (DBES) is a frequently used technique for characterizing the sub-nanometer pore structure of PA membranes, which is in turn related to their crosslinking degree or density⁵. DBES results (**Supplementary Fig. 16**) show that the S parameter for both SNF0-NF0.5 and SNF20-NF0.5 membranes ranged from 0.48 to 0.5 within 1 Kev positron energy (which typically detects the signal from the PA rejection layer⁵). The range of S parameter in our study is comparable to reported in the literature for PA TFC NF membranes (**Supplementary Table 5**).”

Supplementary Fig. 16 | The Doppler broadening energy spectroscopy results of SNF0-NF0.5 and SNF20-NF0.5.

Supplementary Table 5. A comparison of S parameters of DBES between this study and other nanofiltration studies in the literature.

Membrane chemistry	Ranges of S parameter	References (DOI)
Polyamide	0.48 ~ 0.50	This work
Polyamide	0.48 ~ 0.50	10.1038/s41467-020-15771-2 ⁵
Polyamide	0.49 ~ 0.51	1 10.1038/s41467-022-28183-1 ⁶
Polyamide	0.49 ~ 0.51	10.1038/s41467-023-36848-8 ⁷
Polyamide	0.47 ~ 0.49	10.1038/s41467-023-43291-2 ⁸
Polyamide	0.47 ~ 0.49	10.3390/polym12102326 ⁹

B. Molecular weight cut-off approach (MWCO)

To address the reviewer's concern, we have included a new **Supplementary Table 6** and the following discussions in the revised Supporting Information:

(Supporting Information: Page 29, Lines 378-381)

MWCO has been widely used to characterize the pore size property of the PA NF membranes, and MOCO is related to the cross-linking or density of PA. **Supplementary Fig. 29** indicated that the MWCO of SNF20-NF0.1* membranes was 633 Da, which is comparable to many PA TFC NF membranes reported in the literature (**Supplementary Table 6**).

Supplementary Fig. 29 | The MWCO characterization for SNF20-NF0.1* membranes. The presented data is the averaged value based on three distinct samples ($n = 3$).

Supplementary Table 6 | A comparison of MWCO between this study and other polyamide TFC NF membranes reported in the literature.

Membrane chemistry	MWCO (Da)	References (DOI)
Polyamide	633	This work
Polyamide	600 ~ 800	10.3390/w11122512 ¹⁰
Polyamide	500	10.3390/membranes14020038 ¹¹
Polyamide	560	10.1016/j.memsci.2023.122351 ¹²
Polyamide	725.1	10.1016/j.desal.2023.116575 ¹³
Polyamide	579	10.1016/j.memsci.2022.121321 ¹⁴
Polyamide	534	10.1016/j.memsci.2022.121321 ¹⁴
Polyamide	858.9	10.1016/j.memsci.2021.119942 ¹⁵
Polyamide	640.6	10.1016/j.memsci.2021.119942 ¹⁵
Polyamide	514.2	10.1016/j.memsci.2021.119942 ¹⁵
Polyamide	850	10.1016/j.desal.2015.04.027 ¹⁶
Polyamide	800	10.1016/j.memsci.2020.117997 ¹⁷
Polyamide	639	10.1016/j.seppur.2020.118042 ¹⁸
Polyamide	551	10.1016/j.compositesb.2021.108686 ¹⁹
Polyamide	726.7	10.1016/j.memsci.2024.122484 ²⁰

Comment #3. Fig. 4(c)

The SNF-NF membrane showed much higher rejection for Na₂SO₄(96%) than MgSO₄(about 50%), which is far from the literature. In general, MgSO₄ rejection is higher than or similar to Na₂SO₄ rejection. For your reference, a couple of papers are provided below.

[1] N.P. i Hernando, Nanofiltration and hybrid sorption: ultrafiltration processes for improving water quality, in, Universitat Politècnica de Catalunya, 2017.

[2] A.S. Colburn, Synthesis, Functionalization, and Application of Nanofiltration and Composite Membranes for Selective Separations, (2019).

The authors should elaborate on why the SNF-NF membrane exhibited a significantly different tendency in the rejection rates of Na₂SO₄ and MgSO₄ from the literature.

OUR RESPONSE:

The reviewer has raised an interesting point. We would like to adopt Donnan exclusion theory to elaborate on the rejection difference between Na₂SO₄ and MgSO₄. In addition, we have compiled a table from the published literature, which showed many PA-based NF membranes have a higher rejection rate of Na₂SO₄ over MgSO₄.

According to the Donnan exclusion theory, a more negatively charged NF membrane could reject Na₂SO₄ (1:2 salt) better than MgSO₄ (1:1 salt). Our SNF NF membrane exhibited a negatively charged surface at a testing pH of ~ 6, as indicated by the zeta potential results. The rejection of salts can be expressed based on the following equation when Donnan exclusion dominates (DOI: 10.1021/acscami.5b12723 and 10.1016/s1383-5866(98)00070-7)^{21, 22}:

$$R = 1 - \frac{c_i^m}{c_i} = 1 - \left(\frac{|z_i|c_i}{c_x^m + |z_1|c_i^m} \right)^{|z_i/z_j|}$$

Where z_i and z_j are the valence of co-ions and counterions, c_i and c_i^m are the concentrations of co-ions in the bulk solution and membrane respectively, c_x^m is the membrane charge concentration, and subscripts i and j indicate co-ions and counterions, respectively.

Our results agree well with the Donnan exclusion theory and the literature data. To further address the reviewer's concern, we have compiled **Supplementary Table 7**, which lists PA-based NF membranes showing a higher rejection rate of Na₂SO₄ over MgSO₄.

Although our SNF-NF membranes show higher Na₂SO₄ rejection over MgSO₄, the actual solutes rejection of NF membranes is compounded by other factors (e.g., membrane pore size, surface charge, dielectric effect, and hydration energy). Therefore, some NF membranes may demonstrate a higher or similar MgSO₄ rejection than that for Na₂SO₄; Other NF membranes had a lower MgSO₄ rejection than Na₂SO₄. For example, for the reference the reviewer kindly provided (*N.P. i Hernando, Universitat Politècnica de Catalunya, 2017, Chapter 3.2.1, Page 32*)²³, some types of membranes showed higher Na₂SO₄ rejection over MgSO₄, while other membranes demonstrated similar rejections for those two types of salts.

Since this point is beyond the scope of this study, further work could explore the correlation between membrane properties as well as operational conditions and the rejection properties of Na₂SO₄ and MgSO₄.

We have included the following sentences and new **Supplementary Table 7** in the revised Supporting Information:

(Supporting Information: Page 43, Lines 524-534)

NF membranes may have higher Na₂SO₄ rejection compared to that of MgSO₄, often due to the Donnan exclusion effect^{21, 22}. **Supplementary Table 7** listed some PA-based NF membranes with a higher rejection rate for Na₂SO₄ than MgSO₄, which are consistent with our rejection results in this work.

Supplementary Table 7. The Na₂SO₄ and MgSO₄ rejection of polyamide NF membranes

Membrane chemistry	Test Conditions (feed concentration; applied pressure)*	Na ₂ SO ₄ Rejection (%)	MgSO ₄ Rejection (%)	References (DOI)
Polyamide	1000 ppm; 5 bar	99.6	93.2	10.1126/science.adi9531 ²⁴
Polyamide	2000 ppm; 10 bar	96.0	93.4	10.1038/s41467-020-19809-3 ²⁵
Polyamide	1000 ppm, 4 bar	95.3	~90.0	10.1038/s41467-018-04467-3 ²⁶
Polyamide	2000 ppm; 5 bar	94.0	84.8	10.1039/d1ta04763a ²⁷
Polyamide	2000 ppm; 5 bar	98.8	85.5	10.1039/d1ta04763a ²⁷
Polyamide	1000 ppm; 5 bar	95.5	~80.0	10.1016/j.memsci.2020.117971 ²⁸
Polyamide	1500 ppm; 4 bar	97.5	91.2	10.1039/c9ta02299f ²⁹
Polyamide	1000 ppm; 2 bar	92.0	67.6	10.1016/j.memsci.2017.09.016 ³⁰

Polyamide	1000 ppm; 6 bar	99.7	86.0	10.1039/c7ta00501f ³¹
Polyamide	1000 ppm; 5 bar	94.8	72.5	10.1007/s10853-018-2369-2 ³²
Polyamide	1000 ppm; 2 bar	87.6	~71.0	10.1016/j.memsci.2013.12.060 ³³

* The concentrations are the salt concentrations in a single salt separation test; the pressures are the applied hydraulic pressure under cross-flow filtration mode. In addition, all membranes listed in the table are PA TFC NF membranes with a negative surface zeta potential at the tested condition.

Reference

1. Semião AJC, Habimana O, Cao H, Heffernan R, Safari A, Casey E. The importance of laboratory water quality for studying initial bacterial adhesion during NF filtration processes. *Water Research* **47**, 2909-2920 (2013).
2. Thummar UG, Koradiya A, Saxena M, Poliseti V, Ray P, Singh PS. Scaling-up of robust nanofiltration membrane of ultrathin-film-composite structure. *Desalination* **530**, 115650 (2022).
3. Abdullah SZ, Bérubé PR, Horne DJ. SEM imaging of membranes: Importance of sample preparation and imaging parameters. *Journal of Membrane Science* **463**, 113-125 (2014).
4. Xue Y-R, *et al.* Harmonic amide bond density as a game-changer for deciphering the crosslinking puzzle of polyamide. *Nature Communications* **15**, 1539 (2024).
5. Liang Y, *et al.* Polyamide nanofiltration membrane with highly uniform sub-nanometre pores for sub-1 Å precision separation. *Nature Communications* **11**, (2020).
6. Shen L, *et al.* Polyamide-based membranes with structural homogeneity for ultrafast molecular sieving. *Nature Communications* **13**, 500 (2022).
7. Zhao C, *et al.* Polyamide membranes with nanoscale ordered structures for fast permeation and highly selective ion-ion separation. *Nature Communications* **14**, 1112 (2023).
8. Zhao G, Gao H, Qu Z, Fan H, Meng H. Anhydrous interfacial polymerization of sub-1 Å sieving polyamide membrane. *Nature Communications* **14**, 7624 (2023).
9. Ang MBMY, *et al.* Surface Properties, Free Volume, and Performance for Thin-Film Composite Pervaporation Membranes Fabricated through Interfacial Polymerization Involving Different Organic Solvents. *Polymers* **12**, 2326 (2020).
10. Cooray T, *et al.* Drinking-Water Supply for CKDu Affected Areas of Sri Lanka, Using Nanofiltration Membrane Technology: From Laboratory to Practice. *Water* **11**, 2512 (2019).
11. Chidichimo F, Basile MR, Conidi C, De Filipo G, Morelli R, Cassano A. A New Approach for Bioremediation of Olive Mill Wastewaters:

- Combination of Straw Filtration and Nanofiltration. *Membranes (Basel)* **14**, (2024).
12. Zhang Y, *et al.* Based on high cross-linked structure design to fabricate PEI-based nanofiltration membranes for Mg²⁺/Li⁺ separation. *Journal of Membrane Science* **693**, 122351 (2024).
 13. Li T, Zhang X, Zhang Y, Wang J, Wang Z, Zhao S. Nanofiltration membrane comprising structural regulator Cyclen for efficient Li⁺/Mg²⁺ separation. *Desalination* **556**, 116575 (2023).
 14. Li H, Li Y, Li M, Jin Y, Kang G, Cao Y. Improving Mg²⁺/Li⁺ separation performance of polyamide nanofiltration membrane by swelling-embedding-shrinking strategy. *Journal of Membrane Science* **669**, 121321 (2023).
 15. Wu M-B, *et al.* Positively-charged nanofiltration membranes constructed via gas/liquid interfacial polymerization for Mg²⁺/Li⁺ separation. *Journal of Membrane Science* **644**, 119942 (2022).
 16. Li X, Zhang C, Zhang S, Li J, He B, Cui Z. Preparation and characterization of positively charged polyamide composite nanofiltration hollow fiber membrane for lithium and magnesium separation. *Desalination* **369**, 26-36 (2015).
 17. Wu H, *et al.* A novel nanofiltration membrane with [MimAP][Tf2N] ionic liquid for utilization of lithium from brines with high Mg²⁺/Li⁺ ratio. *Journal of Membrane Science* **603**, 117997 (2020).
 18. Xu P, Hong J, Xu Z, Xia H, Ni Q-Q. Novel aminated graphene quantum dots (GQDs-NH₂)-engineered nanofiltration membrane with high Mg²⁺/Li⁺ separation efficiency. *Separation and Purification Technology* **258**, 118042 (2021).
 19. Xu P, Hong J, Xu Z, Xia H, Ni Q-Q. MWCNTs-COOK-assisted high positively charged composite membrane: Accelerating Li⁺ enrichment and Mg²⁺ removal. *Composites Part B: Engineering* **212**, 108686 (2021).
 20. Zha Z, Li T, Hussein I, Wang Y, Zhao S. Aza-crown ether-coupled polyamide nanofiltration membrane for efficient Li⁺/Mg²⁺ separation. *Journal of Membrane Science* **695**, 122484 (2024).
 21. Schaep J, Van der Bruggen B, Vandecasteele C, Wilms D. Influence of ion size and charge in nanofiltration. *Separation and Purification*

Technology **14**, 155-162 (1998).

22. Wang J, *et al.* Graphene Oxide as an Effective Barrier on a Porous Nanofibrous Membrane for Water Treatment. *ACS Applied Materials & Interfaces* **8**, 6211-6218 (2016).
23. i Hernando NP. Nanofiltration and hybrid sorption: ultrafiltration processes for improving water quality.). Universitat Politècnica de Catalunya (2017).
24. Zhang Y, *et al.* Ice-confined synthesis of highly ionized 3D-quasilayered polyamide nanofiltration membranes. *Science* **382**, 202-206 (2023).
25. Yuan B, Zhao S, Hu P, Cui J, Niu QJ. Asymmetric polyamide nanofilms with highly ordered nanovoids for water purification. *Nature Communications* **11**, 6102 (2020).
26. Wang Z, *et al.* Nanoparticle-templated nanofiltration membranes for ultrahigh performance desalination. *Nature Communications* **9**, 2004 (2018).
27. Sarkar P, Modak S, Ray S, Adupa V, Reddy KA, Karan S. Fast water transport through sub-5 nm polyamide nanofilms: the new upper-bound of the permeance–selectivity trade-off in nanofiltration. *J Mater Chem A* **9**, 20714-20724 (2021).
28. Yang S, Wang J, Fang L, Lin H, Liu F, Tang CY. Electrospayed polyamide nanofiltration membrane with intercalated structure for controllable structure manipulation and enhanced separation performance. *Journal of Membrane Science* **602**, 117971 (2020).
29. Zhu J, *et al.* MOF-positioned polyamide membranes with a fishnet-like structure for elevated nanofiltration performance. *J Mater Chem A* **7**, 16313-16322 (2019).
30. Wu M, *et al.* Fabrication of composite nanofiltration membrane by incorporating attapulgite nanorods during interfacial polymerization for high water flux and antifouling property. *Journal of Membrane Science* **544**, 79-87 (2017).
31. Wang J-J, Yang H-C, Wu M-B, Zhang X, Xu Z-K. Nanofiltration membranes with cellulose nanocrystals as an interlayer for unprecedented performance. *J Mater Chem A* **5**, 16289-16295 (2017).

32. Wei C, *et al.* One-step fabrication of recyclable polyimide nanofiltration membranes with high selectivity and performance stability by a phase inversion-based process. *Journal of Materials Science* **53**, 11104-11115 (2018).
33. Li Y, *et al.* Surface fluorination of polyamide nanofiltration membrane for enhanced antifouling property. *Journal of Membrane Science* **455**, 15-23 (2014).

Reviewers' Comments:

Reviewer #3:

Remarks to the Author:

The authors addressed the reviewer's concerns, and the revised work is recommended for publication.